# Pix2Code: Learning to Compose Neural Visual Concepts as Programs

**Antonia Wüst**[1]    **Wolfgang Stammer**[1,2]    **Quentin Delfosse**[1]    **Devendra Singh Dhami**[3]    **Kristian Kersting**[1,2,4,5]

[1]AI and ML Group, TU Darmstadt
[2]Hessian Center for AI (hessian.AI)
[3]Eindhoven University of Technology
[4]Centre for Cognitive Science, TU Darmstadt
[5]German Research Center for AI (DFKI)

## Abstract

The challenge in learning abstract concepts from images in an unsupervised fashion lies in the required integration of visual perception and generalizable relational reasoning. Moreover, the unsupervised nature of this task makes it necessary for human users to be able to understand a model's learned concepts and potentially revise false behaviors. To tackle both the generalizability and interpretability constraints of visual concept learning, we propose Pix2Code, a framework that extends program synthesis to visual relational reasoning by utilizing the abilities of both explicit, compositional symbolic *and* implicit neural representations. This is achieved by retrieving object representations from images and synthesizing relational concepts as $\lambda$-calculus programs. We evaluate the diverse properties of Pix2Code on the challenging reasoning domains, Kandinsky Patterns, and CURI, testing its ability to identify compositional visual concepts that generalize to novel data and concept configurations. Particularly, in stark contrast to neural approaches, we show that Pix2Code's representations remain human interpretable and can easily be revised for improved performance.

## 1 INTRODUCTION

Humans possess the ability to identify recurring concepts in their daily lives, *e.g.*, a driver can identify when pedestrians have the priority independent of the number of pedestrians or other changing properties of the traffic. However, learning such *visual concepts*, particularly without supervision, still poses a major challenge for machine learning (ML) models. This is notably due to the diversity of visual scenes (*cf.* Fig. 1), but also the immense space of possible concepts that can be arbitrarily composed of many subconcepts.

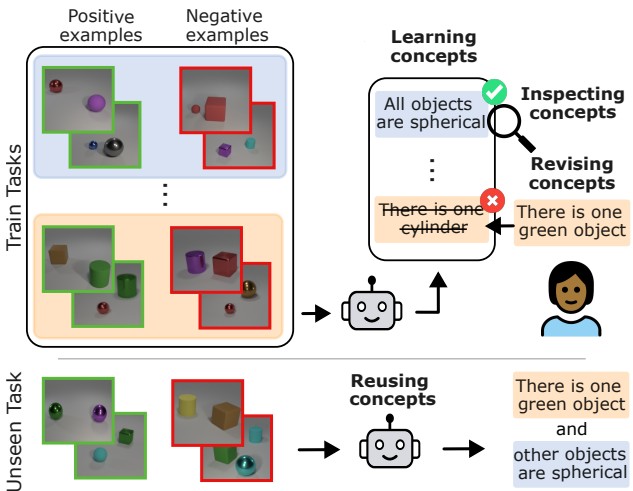

Figure 1: Interpretable visual concept learning: learning concepts from few images that can *generalize* to unseen examples and unseen concepts, such that human users can *inspect* and potentially *revise* suboptimal learned concepts.

Moreover, it remains necessary for human users to be able to inspect the learned concepts and revise potential errors or shortcuts before deployment of such learning systems, particularly in unsupervised learning settings.

Current ML approaches (*e.g.*, [Santoro et al., 2017]) that tackle this challenging task still have issues, *e.g.*, with detecting visual concepts based on object relations as well as generalizing in few-shot learning scenarios (*cf.* [Stabinger et al., 2021] for a survey). [Kim et al., 2018] propose an approach that generalizes to unseen image samples of a concept, but not to unseen concepts. Vedantam et al. [2021], on the other hand, show promising results in terms of generalization to unseen concepts, however, the authors neglect to investigate other forms of generalization, *e.g.*, when the number of objects in an image is increased. Moreover, the nature of the implicit *neural* representations make the learned concepts opaque for human users and impractical to revise.

An orthogonal research field that incorporates generalization, inspectability and revisability for concept learning in general is program synthesis, where knowledge is learned in the form of explicit *programs*. Not only do programs allow to extrapolate to novel, unseen inputs regardless of the number of objects, but their compositional nature is particularly useful for learning to reuse existing knowledge in new ways [Ellis et al., 2021, Stengel-Eskin et al., 2024], particularly in symbolic list processing and text editing settings [Balog et al., 2017, Ellis et al., 2021]. In addition, even the longest programs are *readable* for human users [Cambronero et al., 2023], thereby offering an inherent form of interpretability. Lastly, program synthesis approaches offer easy human revision [Trivedi et al., 2021] such as rewriting or updating the programs. Despite all of this, program synthesis approaches have not been utilized to learn complex *visual* concepts from raw images up to now, likely due to the difficulty of mapping images to symbolic representations.

This work introduces Pix2Code, a neuro-symbolic framework for generalizable, inspectable and revisable visual concept learning. Using both neural and program synthesis components, Pix2Code integrates the power of neural representations with the generalizability and readability of program representations. During inference, Pix2Code extracts symbolic object representations from raw image inputs uses these to synthesize $\lambda$-calculus programs, that serve as concept classifiers (*i.e.*, "Do novel images contain this concept?"), but also as inherent interpretations of these concepts. Pix2Code learns to abstract visual concepts by training both a generative program library and a program recognition model based on wake-sleep learning. In our evaluations, we investigate the advantages of Pix2Code in terms of generalization, *e.g.*, for novel concept combinations, but also extend the evaluation setting of previous works to entity generalization, *i.e.*, generalizing to novel instances of a concept. Lastly, we show that the retrieved concept representations of Pix2Code are inspectable and can easily be revised in case of confounded or suboptimal behavior.[1]

Overall we make the following contributions:

**(i)** We frame visual concept learning in the context of program synthesis in our Pix2Code framework.

**(ii)** Pix2Code learns visual concept representations that are generalizable to unseen concepts.

**(iii)** We effectively revise the learned representations via human guidance to mitigate suboptimal behavior.

**(iv)** We identify limitations with respect to concept generality in the existing concept learning benchmarks and show how this can be alleviated via Pix2Code.

Let us now provide a formal description of our Pix2Code framework, its inference, learning, and revision processes. Next, we move on to experimental evaluations and conclude after presenting related works.

---

[1]Code and data available at github.com/ml-research/pix2code.

# 2 PIX2CODE

In our work, we consider learning *visual concepts*, *i.e.*, general ideas that are fundamental to the understanding of a visual scene (*cf.* Fig. 1). The goal of the Pix2Code framework is to discover such concepts in a generalizable, interpretable, and revisable manner. This is achieved by combining differentiable token-based object representations with program synthesis such that concepts are represented as programs.

Formally, we consider a set of images $X$. For an image, $x \in X$, if a concept, $c$, is appearing in the image, we denote $c \subset x$. Following the setup of Vedantam et al. [2021], we consider the goal of identifying a specific concept, $c$, from a positive subset of $X$, $X^+ := \{x_i^+\}_{i=1}^N$, and a negative set $X^- := \{x_i^-\}_{i=1}^M$. Thus, the goal is to obtain a model, $f_\Theta$ (parameterized by $\Theta$) which proposes a concept, $f_\Theta(X^+, X^-) = c$, that separates $X^+$ from $X^-$, *i.e.*, it must hold that $\forall x_i^+ \in X^+, c \subset x_i^+$ *and* $\forall x_i^- \in X^-, c \not\subset x_i^-$. An overview of how Pix2Code achieves this is shown in Fig. 2. Let us provide a step-by-step description of this, beginning with Pix2Code's inference, followed by its training and revision procedures.

**Concept learning as a program synthesis task.** To obtain visual concepts (*e.g.*, *all objects are spheres*) from an image, the first step of the Pix2Code framework is to cast the above problem of unsupervised concept learning into a suitable program synthesis setting. For that, we recast the initial task (that consists of the tuple of a positive and negative image set, *cf.* Fig. 1), $\{X^+, X^-\}$, to a binary classification task:

$$T := \{(x_i, y_i)\}_{i=1}^{N+M}, \tag{1}$$

where $x_i \in \{X^+\}$ with $y_i = 1$ for $i \in \{0, ..., N\}$, and $x_i \in \{X^-\}$ with $y_i = 0$ for $i \in \{N+1, ..., N+M\}$.

**Transforming images to symbolic object representations.** Visual concepts are based on objects, their attributes and relations. A necessary first step for performing visual concept learning is therefore to identify relevant objects from an image and extract corresponding object representations. Moreover, in order to perform visual concept learning via program synthesis, we specifically require symbolic object representations. Given a pretrained object extraction model, $h_\psi$, Pix2Code extracts a set of discrete representations, $O_i$, from an image $x_i$ that contains $K_i$ objects:

$$O_i := h_\psi(x_i) = \{o_j\}_{j=1}^{K_i}. \tag{2}$$

Each object representation, $o_j \in O_i$, corresponds to a sequence of tokens: $o_j := [x_{\min}, y_{\min}, x_{\max}, y_{\max}, a_1, ..., a_C]$, that includes the object's bounding box coordinates, $[x_{\min}, y_{\min}, x_{\max}, y_{\max}] \in \mathbb{N}^4$, as well as relevant object properties $[a_1, ..., a_C] \in \mathbb{N}^C$. For notation reasons, we here consider that all objects in our images possess attributes from the same amount of categories, $C \in \mathbb{N}$, *e.g.*, size, shape, color, and material for the objects of Fig. 1. Moreover, each

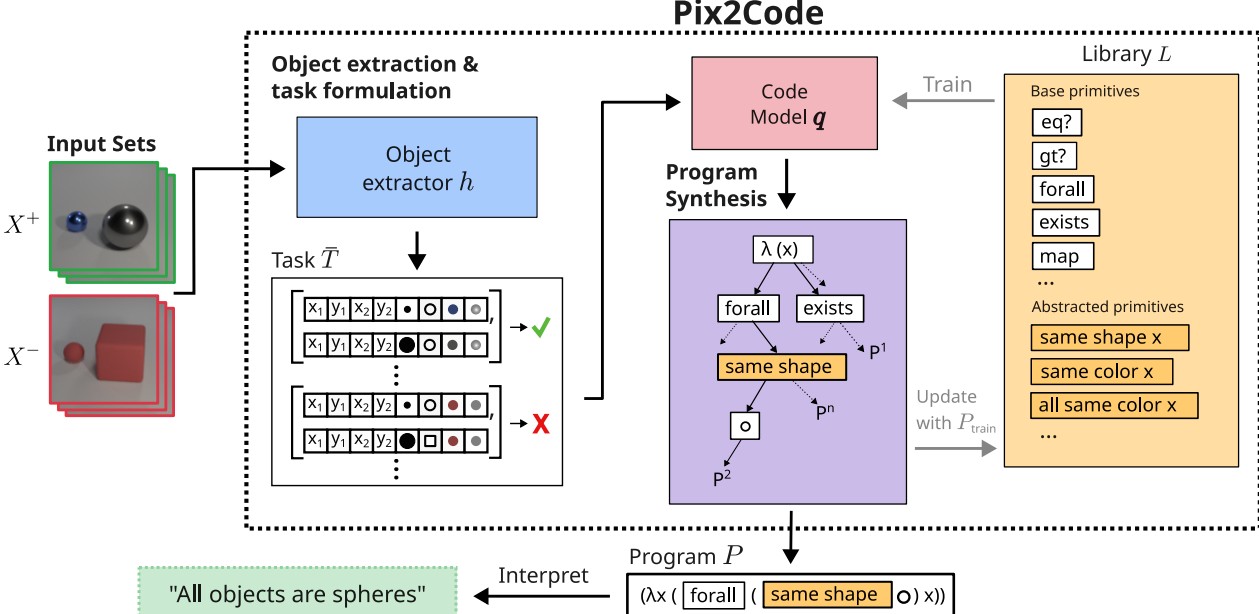

**Figure 2: The Pix2Code architecture.** Objects with bounding box and attribute information are extracted from each positive and negative image example of a visual concept. These representations are converted into a binary classification formulation. The program synthesis component searches for programs to solve each task. This search is based on a probabilistic library that is learned and enhanced during training by frequently used program parts. The result of the search is the visual concept of the image in form of an executable program that can be translated into a corresponding natural language statement.

category contains a finite number of possible attribute instantiations, *i.e.*, $\forall k \in [1, ..., C], a_k \in \{1, ..., d_k\}$ with $d_k \in \mathbb{N}$, *e.g.*, *sphere*, *cube* and *cylinder* for the shape category or *red*, *blue*, *green*, *yellow*, etc. for the color category. Conclusively, the object extractor, $h$, extracts a set of symbolic object representations. The original input is transformed via the two previous steps to obtain the following task representation:

$$\bar{T} := h_\psi(X^+, X^-) = \{(O_i, y_i)\}_{i=1}^{N+M}. \quad (3)$$

**Synthesizing programs from object representations.**
Having obtained symbolic representations of the input task, we now move on to learning abstract programs via Pix2Code's program synthesis module, $g_{L,\phi}$. This consists of two components: a library of learned primitives, $L$, and a code model, $q_\phi$, which predicts the most likely primitives of $L$ given a task. For describing the inference procedure, we consider that $L$ and $q_\phi$ result from an already trained framework. Specifically, $L$ contains base primitives (*e.g.*, *forall*, *eq?*) as well as learned program *primitives* (*e.g.*, *same shape*), represented as reusable functional programs $L := [p_0, ..., p_B]$. Each primitive, $p_j$, possesses a specific arity $m_j \in \mathbb{N}$ (*i.e.*, number of input variables). With $L$ as vocabulary, the code model $q_\phi$ proposes the most likely primitives given a task $\bar{T}$. Thus, during an enumerative search, $q_\phi$ is used to synthesize the most likely program $P$ which encodes the concept separating the examples of $\bar{T}$.

During the enumerative search, programs are constructed by sampling primitives from $q_\phi$. For an efficient search, beam search is used in order to extend only the most likely partial programs under $q_\phi$. At step $\tau$, we denote the concept (in form of an incomplete program) as $P_\tau$ and the primitive selected to extend it as $p_\tau^*$. Specifically, given task $\bar{T}$ and the current partial program $P_{\tau-1}$, the code model $q_\phi$ provides a distribution over the next primitives:

$$\forall p_j \in L : q_\phi(\bar{T}, P_{\tau-1}, p_j) = \rho_\tau(p_j) \in [0, 1]. \quad (4)$$

From this distribution, $p_\tau^*$ is sampled and added to the current program, *i.e.*, $P_\tau = P_{\tau-1} \otimes p_\tau^*$, where $\otimes$ corresponds to placing $p_\tau^*$ within $P_\tau$. The search starts with an initial program as an empty set, *i.e.*, $P_0 = \{\}$. A partial program is complete when the variables of each $p_\tau^*$ are set, *i.e.*, each variable of $p_\tau^*$ has been substituted with a primitive of arity zero or the variable itself is a primitive with set variables. This final step is denoted as $\hat{\tau}$. Finally, at the end of the search the most likely program $P_{\hat{\tau}}$ is obtained, such that

$$g_{L,\phi}(\bar{T}) = P_{\hat{\tau}} =: P. \quad (5)$$

The resulting program is a composition of primitives, *i.e.*, $P = p_0^* \otimes p_1^* \otimes ... \otimes p_{\hat{\tau}}^*$. In the example illustrated in Fig. 2, the retrieved program is $P = \lambda (x) \otimes \texttt{forall} \otimes \texttt{same\_shape} \otimes \texttt{sphere} \otimes x$, which checks whether all objects in the image have a *spherical shape*. Conclusively, the overall inference procedure of Pix2Code is:

$$c := P = g_{L,\phi}(h_\psi(X^+, X^-)). \quad (6)$$

Lastly, we note that the final concept, represented as a program $P$, serves two important purposes. $P$ can be used to (i) classify unseen images (*cf.* Suppl. A and Fig. 7) and at the same time to (ii) provide a transparent procedure of this classification, thus directly serving as an explanation.

**Learning programs from images.** To train Pix2Code, we need to optimize each of its parameters $\Theta := \{\psi, L, \phi\}$. In our evaluations, we differentiate between optimizing the parameter set of the object extractor, $\psi$, and jointly optimizing the library $L$ and parameter set of the code model, $\phi$, which both represent parameters of the program synthesis model. The training of the object extractor is independent and can, in principle, be done in an unsupervised manner [Delfosse et al., 2023c]. We here follow the procedure of Chen et al. [2022]. However, instead of detecting one class per object as in the original work, we detect $C$ classes per object (one for each attribute category). Specifically, given a training image $x$ with $K$ objects and $C$ attribute categories, the corresponding object sequences are $\hat{y} := \{\hat{o}_j\}_{j=1}^K$, with $\hat{o}_j = (\hat{x}_{\min}^j, \hat{y}_{\min}^j, \hat{x}_{\max}^j, \hat{y}_{\max}^j, \hat{a}_1^j, ..., \hat{a}_C^j)$. The object extractor, $h$, is trained to optimize the maximum-likelihood loss ($\operatorname{argmax}_\psi LL(\hat{y}, h_\psi(x))$) via gradient descent. In this way, $h$ is optimized to identify multiple attributes per object. Further details are provided in Suppl. A.

On the other hand, the library and the code model of the program synthesis component are jointly optimized based on the probabilistic approach of Ellis et al. [2023] and the wake-sleep algorithm of Hinton et al. [1995]. Specifically, $L$ and $q_\phi$ bootstrap each other. $L$ initially contains only base primitives, *i.e.*, the domain specific language (DSL), and is parametrized by $\mu$, that corresponds to the prior probability of each primitive (initialized uniformly). For a training task $\bar{T}_i \in \bar{T}_{\text{train}}$, a set $\Pi^i = \{P_s^i\}_{s=1}^S$ of programs is sampled from $L_\mu$ (wake phase), where $S \in \mathbb{N}$ denotes the number of maximum considered programs per task. The retrieved programs are used to train $q_\phi$ (sleep phase), following:

$$\mathcal{L} = \mathbb{E}_{\bar{T}_i \sim \bar{T}_{\text{train}}} \left[ \log q_\phi (\operatorname*{arg\,max}_{P_s^i \in \Pi^i} p(P_s^i \mid \bar{T}_i, L_\mu)) \right]. \quad (7)$$

Moreover, Pix2Code uses a dreaming phase in which new task-program pairs are created, *i.e.*, new object-centric data and programs are generated to additionally train $q_\phi$ following Eq. 7. $L_\mu$ is optimized using programs sampled from the updated $q_\phi$. For this, $S$ programs for each task are sampled via $q$: $P_{\text{train}} = \{q_\phi(\bar{T}_i) | \forall \bar{T}_i \in \bar{T}_{\text{train}}\}$. With this set of sampled programs, the probability of the library primitives are updated via maximum a posteriori estimation. Further, frequently used program parts within $P_{\text{train}}$ are identified and added to $L$ to improve its objective function (*cf.* Suppl. A and Ellis et al. [2023] for further details).

**Revising latent concept representations.** Pix2Code integrates interpretable and accessible components and latent representations that allow human users to identify and revise potentially suboptimal model behavior (*e.g.*, overfitting, confounding [Schramowski et al., 2020] or other forms of shortcut learning [Geirhos et al., 2020]).

This work mainly differentiates between the following revision possibilities: (i) removing possibly undesirable primitives from $L$, (ii) adding relevant, yet previously undiscovered primitives to $L$ and (iii) modifying existing primitives in $L$. This last form of revision can be further subdivided into (iii-a) modifying the explicit program representation of a primitive or (iii-b) finetuning $q_\phi$ to reweight the probabilities of specific primitives in $L$, *e.g.*, via one of the loss-based approaches of eXplanatory Interactive Learning (XIL) [Schramowski et al., 2020, Friedrich et al., 2023].

# 3 EXPERIMENTAL EVALUATION

In our experimental evaluations, we show how Pix2Code uses programs to discover complex visual patterns from few examples in an interpretable and revisable manner. Overall, our evaluations aim to answer the following questions:

**(Q1)** Is Pix2Code able to learn abstract visual concepts?

**(Q2)** Can Pix2Code learn concepts that generalize to unseen combinations of concept components?

**(Q3)** Can these concepts generalize to inputs with unseen number of objects?

**(Q4)** Are the concept representations interpretable?

**(Q5)** Can Pix2Code be revised to correct for suboptimal behavior?

**(Q6)** Can Pix2Code abstract concepts from real-world data?

## 3.1 EXPERIMENTAL SETUP

We here provide setup details to allow for reproducibility.

**Data.** For evaluating Pix2Code, we create an extensive dataset from the **Kandinsky Patterns** framework [Holzinger et al., 2019] called **RelKP**, that contains images of 2D objects (depicted in Fig. 8), with the attributes *color*, *shape* and *size*. The images embed patterns such as "there are two pairs of objects with the same shape", similar to Shindo et al. [2023a] (for further details, *cf.* Suppl. C.1). We further use the **CURI** dataset [Vedantam et al., 2021], containing images of 3D objects (illustrated in Fig. 1), with the attributes *color*, *shape*, *size* and *material*. CURI is designed to test compositional generalization. It contains 8 different concept splits, which are based on specific properties. For each split, concepts with these properties occur only in the test sets and not in the training sets (*cf.* Suppl. C.2 for more details). For both datasets, the images are grouped by abstract visual concepts, which are based on the objects' attributes and relations between them. For each concept there

is a *task* that contains at least one support and one query set, each holding positive and negative image examples of the concept. The objective is to recognize the underlying concept from the support set and, based on that, classify the examples from the query set correctly. The datasets contain training tasks and held-out test tasks. To reduce the computational burden, we randomly select a subset of 100 training concepts from each CURI split; however, we evaluate on the full, original test concepts of each split. For investigating entity generalization (Q3) and confounding behavior (Q5), we introduce extensions of CURI. These contain images created via the data generator framework of Stammer et al. [2021] (*cf.* Suppl. D.5 and *cf.* Suppl. D.7). Finally, to evaluate real-world concepts we created a small set of abstract concepts based on the popular MS COCO dataset [Lin et al., 2014] (*cf.* Suppl. D.9).

**Models.** In our evaluations, we compare the performance of our neuro-symbolic Pix2Code approach to the purely neural model of Vedantam et al. [2021], here referred to as *CURI-B*, and provide further details in Suppl. B. The model of Vedantam et al. [2021] was introduced with 4 different pooling alternatives. We report performances of the best performing alternative (*cf.* Tab. 9 for a detailed comparison). For Pix2Code, we base the pretrained object extraction on the approach of Chen et al. [2022] to transform the input images into sequences of natural numbers (representing the objects and their attributes (*cf.* Tab. 6)) and the program synthesis component on the approach of Ellis et al. [2023]. We utilize a domain specific language (DSL) that operates on the specific object representations, and that contains base program primitives, *e.g.*, functions like *forall* and logical operators *and* and *or* (*cf.* Tab. 7 and Suppl. A for details). The program synthesis component is pretrained on the ground truth object representations (denoted as *schema* input). However, unless noted otherwise, it receives the neurally extracted object representations during evaluations. Notably, whereas CURI-B must be optimized on both the support and query examples of a task, Pix2Code is only optimized based on the support examples.

**Metrics.** We evaluate both models' accuracies on the query sets of the test tasks, each averaged over 3 seeded reruns. Since RelKP and CURI contain more negative examples, we provide class balanced accuracies (*i.e.*, mean between accuracies on the positive set and the negative one) over all test tasks. Since Pix2Code uses an enumerative search to retrieve programs that solve test tasks, it may occur that no program is found that solves the task within the preset search time. In this case, we assume a random accuracy for the corresponding test task (*i.e.*, $50\%$), to appropriately compare to the neural baseline model (which always produces an output). We thus differentiate between the class accuracy with random guessing for not found programs, denoted as "Acc@all", from the mean accuracy specifically only of the found programs, denoted as "Acc@solved".

Table 1: Mean test accuracy on Kandinsky and CURI concepts with iid train test splits.

| Dataset | CURI-B | Pix2Code Acc@all | Pix2Code Acc@solved |
|---|---|---|---|
| RelKP | 59.69 $\pm 0.83$ | **90.05** $\pm 0.80$ | 92.93 $\pm 0.98$ |
| CURI (iid) | 66.68 $\pm 1.50$ | **71.54** $\pm 1.15$ | 81.75 $\pm 3.12$ |

Table 2: Mean accuracy (with std) for meta-test tasks of CURI splits reported individually and as the median (with median absolute deviation) over all splits.

| CURI (Splits) | CURI-B | Pix2Code Acc@all | Pix2Code Acc@solved |
|---|---|---|---|
| Boolean | 67.86 $\pm 1.21$ | **78.93** $\pm 1.14$ | 91.05 $\pm 2.33$ |
| Counting | **62.19** $\pm 2.44$ | 55.52 $\pm 2.14$ | 65.73 $\pm 2.19$ |
| Extrinsic | 72.56 $\pm 0.40$ | **78.31** $\pm 1.60$ | 88.23 $\pm 1.70$ |
| Intrinsic | 67.85 $\pm 2.50$ | **87.35** $\pm 3.21$ | 92.09 $\pm 0.40$ |
| Bind.(color) | 69.89 $\pm 1.54$ | **79.03** $\pm 2.26$ | 87.14 $\pm 2.27$ |
| Composition | 67.63 $\pm 0.53$ | **74.82** $\pm 0.10$ | 86.51 $\pm 0.98$ |
| Bind.(shape) | 66.35 $\pm 0.36$ | **74.37** $\pm 2.40$ | 87.14 $\pm 2.27$ |
| Complexity | 65.24 $\pm 0.14$ | **72.37** $\pm 0.51$ | 77.43 $\pm 0.35$ |
| Median | 67.74 $\pm 0.87$ | **76.57** $\pm 1.87$ | 87.14 $\pm 1.95$ |

## 3.2 EVALUATIONS

Let us now verify if Pix2Code can learn abstract visual interpretable concept representations, useful for solving logically challenging tasks from both synthetic and real world data, and can easily generalize to novel situations or be revised.

**Learning visual concepts (Q1).** We first investigate whether Pix2Code can learn visual concepts that allow to separate positive from negative images. Specifically, we evaluate Pix2Code and the (neural) CURI-B algorithm on the RelKP and CURI datasets. For these evaluations, we assume independent and identically distributed (iid) training and test task sets. This stands in contrast to more structured, curriculum-like task splits of later evaluations. Focusing first on the two left columns of Tab. 1, we observe that Pix2Code largely outperforms the neural baseline over both datasets, even when assuming random performance for test tasks for which no program is found. The accuracy of only the found test tasks ($80.93\%$ solved tasks over both datasets *cf.* Tab. 13) in the right-most column of Tab. 1 is significantly higher on every task. Thus, when Pix2Code discovers relevant concepts these provide considerably improved generalization to unseen image samples of the same task, which motivates for further investigation on the visual concepts generalization. We refer to Fig. 6 for visualization of Pix2Code's learned concept library. Overall, our results indicate that visual concept learning via our program synthesis based Pix2Code represents a competitive alternative to purely neural based approaches (*cf.* Tab. 12 for ablations).

Table 3: Class balanced accuracy on AllCubes-$N$ and AllMetalOneGray-$N$ for CURI-B and Pix2Code.

| Dataset | CURI-B | Pix2Code |
|---|---|---|
| AllCubes-CURI | 77.19 $\pm 6.56$ | **100.00** $\pm 0.00$ |
| AllCubes-5 | 70.33 $\pm 5.25$ | **100.00** $\pm 0.00$ |
| AllCubes-8 | 57.83 $\pm 6.20$ | **100.00** $\pm 0.00$ |
| AllCubes-10 | 56.00 $\pm 5.61$ | **100.00** $\pm 0.00$ |
| AllMetalOneGray-CURI | 60.00 $\pm 3.33$ | **96.94** $\pm 4.32$ |
| AllMetalOneGray-5 | 52.50 $\pm 3.54$ | **100.00** $\pm 0.00$ |
| AllMetalOneGray-8 | 52.17 $\pm 3.06$ | **89.00** $\pm 15.56$ |
| AllMetalOneGray-10 | 54.17 $\pm 4.25$ | **89.17** $\pm 15.32$ |

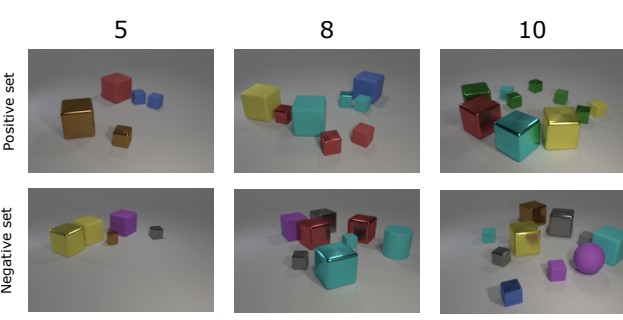

Figure 3: Test examples of AllCubes-5 (left), AllCubes-8 (middle) and AllCubes-10 (right) sets. Positive images contain only cubes, while negative images possess all cubes but one cylinder or sphere.

In the following, we focus on the generalization performances of the evaluated models. We distinguish between two forms of generalization: the ability of a model to reuse previously acquired concepts for composing novel concepts (denoted as *compositional generalization*) and the ability of an extracted representation to generalize to an unseen number of objects (denoted as *entity generalization*). While the first focuses more on the generalizability of the learning components, the second focuses on the generalizability of the concept representations themselves. Let us begin by investigating compositional generalization.

**Generalizing to novel combinations of known concepts (Q2).** We focus these evaluations on the 8 original *compositional* concept splits of the CURI dataset (in contrast to the previous iid splits), which were specifically designed for investigating compositional generalization in concept learning. We provide both models' accuracies in Tab. 2, of each concept split individually, and the median performance over all splits. We observe that in 7 out of 8 CURI splits, Pix2Code greatly outperforms the CURI-B model in terms of generalizing to unseen combinations of concepts (also seen in the median accuracy over all splits). Notably, the counting split appears to be more challenging for Pix2Code, but this can easily be revised, as we show in later evaluations. A possible reason for the overall performance ascendancy of our approach is that CURI-B must learn an individual concept representation for each novel composition, while the modular nature of Pix2Code's programs allows for easy combinations of existing knowledge to form novel representations. Conclusively, we answer Q2 affirmatively and conclude that Pix2Code possess better generalization abilities to unseen concept compositions over the neural baseline (*cf.* Tab. 15 for ablations).

**Generalizing to variable number of objects (Q3).** Let us now move on to the second form of generalizabilty, *i.e.*, entity generalization. While the setup of CURI is valuable for testing the compositional generalization ability of a model, it is insufficient for testing the entity generalizability of its learned concept representations. To investigate Q3, we first need to extend the initial CURI dataset accordingly.

To this end, we select 2 arbitrary concepts from CURI's original tasks, "all objects are cubes" and "all objects are metal and one is gray", and increase the number of objects in the corresponding test images[2] and investigate how well a model can classify these as (still) representing the original concept. Specifically, we create 3 data sets for each of these two concepts with respectively 5, 8 and 10 objects in the test scenes. We refer to these data sets as AllCubes-$N$ and AllMetalOneGray-$N$, where $N \in \{5, 8, 10\}$, and to the original CURI images with "-CURI". We provide example images of AllCubes-$N$ in Fig. 3. Each dataset contains 100 positive image examples and 100 negative ones. We refer to these CURI variations as **CURI-EG**.

As these investigations focus on concept generalization, we revert to using CURI-B and Pix2Code models that were trained with schema input to avoid image encoding noise. These models are trained on the CURI iid split (*cf.* Tab. 17) and evaluated on the CURI-EG test sets. Our results, provided in Tab. 3, show that even with few more objects in the test data set the accuracy of CURI-B drops significantly. In contrast, the performance of Pix2Code stays solid at 100% for the AllCubes-$N$ sets. For every AllMetalOneGray-$N$, Pix2Code significantly outperforms CURI, though in 1 out of 3 seeds Pix2Code did not find the "perfect" concept representation, but one that is overfitting to the support images. This illustrates that programs alone do not overcome suboptimal behaviors such as overfitting. In the next evaluations, we will focus on identifying and revising such behavior.

Overall, these evaluations indicate that the concept representations of Pix2Code can generalize well to inputs with unseen number of objects, answering Q3 affirmatively. The results particularly highlight the importance of entity generalization as a relevant aspect of concept learning validation that neither the evaluations of Tab. 1 nor Tab. 2 could reveal. Moreover, CURI-B's performance in Tab. 3 raises questions concerning the generalizability of its concepts.

---

[2]The original CURI images contain between 2 and 5 objects for both the training and test splits.

Table 4: **Pix2Code's concepts are transparent programs and can be brought to natural language by LLMs.** Examples of original CURI concepts (left), corresponding Pix2Code's program representations (middle) and (right) natural language translations from an LLM (here gpt-4-turbo). We have grouped related concepts and provide syntax highlighting for easier comparison. All programs achieve 100% accuracy on the CURI test scenes.

| Original Concept | Retrieved Program | GPT4 Natural Language Translation |
|---|---|---|
| **All** objects are **cyan** | (λ (x) ((λ (y z) (**forall** (λ (u) (eq? (index y u) z)))) **color cyan** x)) | "All objects are cyan in color." |
| **All** objects are **cubes** | (λ (x) ((λ (y z) (**forall** (λ (u) (eq? (index y u) z)))) **shape cube** x)) | "All objects are cubes." |
| **All** objects are **purple and all** objects are **spheres** | (λ (x) (**and** ((λ (y z) (**forall** (λ (u) (eq? (index y u) z)))) **shape sphere** x) ((λ (v w) (**forall** (λ (a) (eq? (index v a) w)))) **color purple** x))) | "All objects are spheres and all objects are purple." |
| **All** objects are **brown or all** objects are **cubes** | (λ (x) (**or** ((λ (y z) (**forall** (λ (u) (eq? (index y u) z)))) **shape cube** x) ((λ (v w) (**forall** (λ (a) (eq? (index v a) w)))) **color brown** x))) | "All objects are either cubes or all objects are brown." |
| **All** objects are **small and there exists** a **purple** object | (λ (x) (**and** ((λ (y z) (**forall** (λ (u) (eq? (index y u) z)))) **size small** x) (**exists** (λ (v) ((λ (w a b) (eq? (index b w) a)) v **purple color**)) x))) | "All objects are small in size, and there is at least one purple object." |
| There are **three gray** objects | (λ (x) (**eq?** ((λ (y) (**count** (map (λ (z) ((λ (u v) (index u v)) **color** z)) y))) x **gray**) **3**)) | "There are three objects that are gray." |
| There exists an arbitrary object and **there exist three** other objects that are **blue** | (λ (x) (**gt?** ((λ (y) (**count** (map (λ (z) ((λ (u v) (index u v)) **color** z)) y))) x **blue**) **2**)) | "There are more than two objects that are blue in color." |

**Interpreting Pix2Code's concept representations (Q4).** Although our previous results suggest that Pix2Code's programs are more generalizable, Pix2Code can provide suboptimal programs (*cf.* results on AllMetalOneGray) when overfitting or learning shortcuts. However, a considerable advantage of Pix2Code is the readable nature of its program representations, which allows human users to understand and thus detect such suboptimal behaviors. This stands in stark contrast to the opaque concept representations of purely neural approaches such as CURI-B.

We exhibit Pix2Code's transparency in Tab. 4, where we present program solutions for a collection of test tasks from the CURI dataset. The leftmost column describes the target underlying concepts (with increasing complexity over the rows). The middle column presents Pix2Code's corresponding concept representations. Although these programs are written as λ-calculus (which may appear difficult to discern for novices), they possess a straightforward reading procedure with definite variables and operations semantics (provided in Tab. 7). For example, the first program of Tab. 4 reads as follows: the program takes an input list of objects x and applies the function λ (y z), parameterized by *color* and *cyan*, to x. This function applies *forall* with a predicate function λ (u) on each object. The specific predicate function tests if the color attribute of the object representation equals cyan. Overall, the program returns *true* if and only if all objects in x are of color cyan.

Furthermore, large language models can help to translate Pix2Code's λ-calculus programs into natural language statements. We exemplify this in the right-most column of Tab. 4 based on gpt-4-turbo [OpenAI, 2023] (*cf.* Suppl. D.6 for prompting details). In principle, other LLMs can be used as well (as we show in Tab. 18). These can help human users to further understand Pix2Code's proposed concepts. Thus, although the λ-calculus structure of Pix2Code's program representations can present a challenge for novice λ-calculus users, they present a readable and executable knowledge representation and can be translated by LLMs. This provides an affirmative answer to Q4.

**Mitigating Confounders (Q5).** Once a user has identified suboptimal behavior in an AI model, it remains important that they can revise it [Teso and Kersting, 2019, Schramowski et al., 2020], *e.g.*, to ensure trust between model and human. In our last evaluations, we investigate the revisability of Pix2Code's representations and showcase the first two revision forms of Sec. 2: (i) removing primitives from and (ii) adding primitives to Pix2Code's library.

Pix2Code can be affected by shortcut learning. This is exemplified by the last concept of Tab. 4. For this task, there is no negative example that contains *only* 3 blue objects, and thus the retrieved program obtains a perfect accuracy while diverging from the intended concept. Further, "Confounding" can occur when unknown spurious correlations, absent from the query images, appear in the support set images. In this

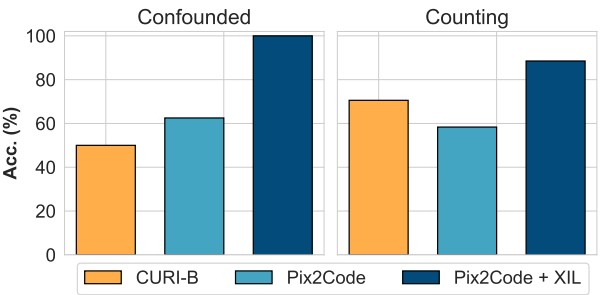

Figure 4: Class balanced accuracies after revising Pix2Code by removing suboptimal primitives from $L$ on the confounded CURI-Hans set (left) and by adding helpful primitives to $L$ in the counting split of CURI (right). Pix2Code +XIL indicates the revised models.

case, Pix2Code's detected programs might classify the support images based on these features and thus fail to apply to the unconfounded query data. We demonstrate this via a confounded task $T_{\mathrm{conf}}$, using the concept "all objects are metal and there exists one cube". In the support set, all objects are cyan (confounding feature), irrespective of the actual underlying concept. In the query set, however, objects possess varying colors. We train Pix2Code on a set of 8 original tasks from the CURI dataset and evaluate on such a confounded test task (*cf.* Suppl. D for details). We refer to this data split as **CURI-Hans** and provide test query accuracy results of Pix2Code and CURI-B in Fig. 4 (left). Both approaches are strongly influenced by the confounder, as indicated by the low query set accuracy (in contrast to the CURI iid base accuracy of Tab. 12), though for Pix2Code this effect is slightly reduced. We can now, however, easily mitigate the behavior of the Pix2Code model by removing the library primitives for color and cyan, as well as abstracted functions that use these. The revised model ("Pix2Code +XIL") reaches 100% test accuracy. It thus appears to ignore the confounder and capture the true concept.

**Revising Pix2Code to Count (Q5).** Another possibility of revising Pix2Code's representations is via the addition of relevant library elements. For example, Pix2Code could lack some relevant DSL primitives such that it can only find shortcut based programs for some concepts. Upon inspection of Pix2Code's concept representations, a user may, however, identify the missing concepts and add these. To test this scenario, we revert to the *Counting* split of CURI, for which Pix2Code had obtained low test accuracy (*cf.* Tab. 2 and Tab. 15). As the concepts from this test set all contain some form of counting operations the low test accuracy indicates that Pix2Code was not able to properly capture the basic concept of "counting". We, therefore, formulate program primitives that count the number of occurrences for each existing attribute (*cf.* Suppl. D for details) and add these primitives to the library of the trained Pix2Code model of *Counting*. The test accuracy of the revised model (Pix2Code

"Exists person and exists dog"

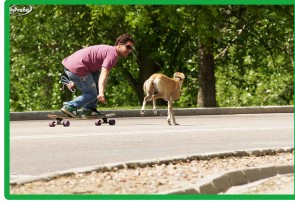 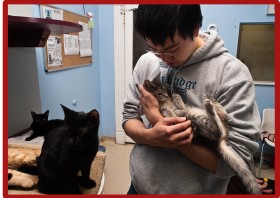

positive example      negative example

Figure 5: Examples of the concept "Exists person and exists dog" based on the COCO dataset.

Table 5: Accuracy of programs synthesized by Pix2Code for concepts in the MS COCO dataset. For each concept, 25 example images were provided and the programs were evaluated on 100 test images.

| COCO Concept | Pix2Code |
|---|---|
| Exists dog | 0.8938 |
| Exists cat | 0.9211 |
| Exists dog and exists person | 0.9286 |
| Exists dog or cat | 0.9324 |
| There are 3 persons | 0.6813 |

+ XIL) nearly jumps to 90% in Fig. 4 (right). Conclusively, Pix2Code allows for easy revision of its programs to overcome suboptimal behavior, answering Q5 affirmatively.

**Extending Pix2Code to Real-World Images (Q6).** In our previous evaluations, we observed that Pix2Code performs impressively at detecting abstract concepts in synthetic data sets. In our last evaluation, we investigate Pix2Code's potential for discovering abstract concepts also in real-world image scenarios. Applying Pix2Code to this setting requires that the object extractor has to detect more visually complex objects and the program synthesis module has to find patterns in more complex input representations (more details in Suppl. D.9). We illustrate this setting based on concepts that are contained in the MS COCO dataset [Lin et al., 2014], *e.g.*, concepts like "There exists a dog" and "There exists a dog or there exists a cat" (*cf.* Suppl. D.9). In Tab. 5 we provide the accuracy of Pix2Code's learned concepts. We observe that Pix2Code's learned programs result in high test set accuracies suggesting that indeed Pix2Code is able to synthesize programs for these real-world concepts. We further refer to Suppl. D.9 for additional discussions, *e.g.*, the influence of noisy object perception. Overall, these results suggest that Pix2Code can abstract concepts from real-world images. We thus answer Q6 affirmatively.

In summary, our evaluations provide evidence of the advantages of utilizing the power of program synthesis for visual concept learning via Pix2Code, in terms of generalizability, interpretability, and revisability.

# 4 RELATED WORK

Pix2Code is closely related to several lines of research, among which program synthesis and concept extraction from images are the closest.

**Program Synthesis.** There has been a recent interest in the task of program synthesis within the realm of machine learning [Chen et al., 2018, Nye et al., 2020, Odena et al., 2021]. Program synthesis has been looked at from various points of view such as neuro-symbolic AI [Parisotto et al., 2017, Bhatia et al., 2018], lifelong learning [Valkov et al., 2018] and interactive machine learning [Zhang et al., 2020, Ferdowsifard et al., 2021]. The various application domains for program synthesis include videos [Sun et al., 2018, Le Moing et al., 2021], images [Laich et al., 2020, Ellis et al., 2021] and text [Ellis et al., 2019, Desai et al., 2016]. There are several methods that develop program synthesis libraries focused on visual reasoning, such as LILO [Grand et al., 2023], ROAP [Tang and Ellis, 2023] and DreamCoder [Ellis et al., 2023]. The biggest drawback of these approaches is the lack of generalization and the process of revision of the learned concepts which Pix2Code addresses.

**Interpretable (Relational) Concepts Learned from Images.** The Neuro-Symbolic Concept Learner of Mao et al. [2019] learns visual concepts from images without any explicit supervision. Whereas Stammer et al. [2022] learns single object-based visual concepts via weak supervision. Lime-Aleph [Rabold et al., 2020] combines the explainable AI method of Lime [Ribeiro et al., 2016] with the classical inductive logic programming system Aleph [Srinivasan, 2001]. The method learns explainable relational concepts on the blocksworld domain. Recently, Shindo et al. [2023a] proposed $\alpha$ILP, a neuro-symbolic framework that can learn generalized rules from complex visual scenes. The advantage of $\alpha$ILP is that it uses differentiable inductive language programming where the logic programs are learned using gradient descent. This was further extended in NEUMANN [Shindo et al., 2023b] where a graph-based differentiable forward reasoner is used for more efficient reasoning framework.

Delfosse et al. [2023b] integrate $\alpha$-ILP in reinforcement learning agents. The value of interpretable relational concepts was further evidenced in the context of reinforcement learning by Delfosse et al. [2024], though both works require prior relational functions. Such interpretable RL agents can also be translated into tree programs [Kohler et al., 2024]. However, these RL applications do not learn to extract concepts, but assume their extraction (using *e.g.* OCAtari [Delfosse et al., 2023a]). Instead, one could integrate Pix2Code into concept bottleneck RL agents. Lastly, [Webb et al., 2023, Kerg et al., 2022, Vaishnav and Serre, 2023] focus on purely neural object-centric approaches for learning relational concepts, which, however, lack the ability to inspect and revise the model's concepts.

# 5 CONCLUSION

In this work, we propose Pix2Code, a neuro-symbolic framework for generalizable, inspectable, and revisable visual concept learning. It captures and reuses concepts as program primitives to compose new concepts, thereby making Pix2Code generalizable to unseen tasks. Our evaluations show that Pix2Code's generalization is especially effective when the number of objects in the visual scene increases, in stark contrast to the neural baseline. Moreover, we show empirically that Pix2Code's learned concepts are interpretable and can be revised via human guidance.

$\lambda$-calculus programs are interpretable but not very natural to humans and can become quite nested for more complex programs. Handling this via a more general program synthesis framework is a natural next step. Integrating the natural language interpretations as part of the training procedure by labeling the learned library primitives with semantic descriptions is another important direction, but also further, pursue applying Pix2Code to more natural images and relations. Additionally, in our evaluations, we have focused on the first two revision procedures of Pix2Code as these represent the more fundamental interactions that a user can perform. We suspect the third type of revision procedure should be a straightforward combination of the two investigated ones or, otherwise, an application of standard XIL approaches. However, future investigations should confirm this. Finally, making the program synthesis component less dependent on the quality of the extracted object representations by allowing probabilistic inputs for the programs can make the Pix2Code framework more widely applicable.

### Acknowledgements

This work was supported by the Priority Program (SPP) 2422 in the subproject "Optimization of active surface design of high-speed progressive tools using machine and deep learning algorithms" funded by the German Research Foundation (DFG). Further, it was supported by the German Federal Ministry of Education and Research and the Hessian Ministry of Higher Education, Research, Science and the Arts (HMWK) within their joint support of the National Research Center for Applied Cybersecurity ATHENE, via the "SenPai: XReLeaS" project, the EU ICT-48 Network of AI Research Excellence Center "TAILOR" (EU Horizon 2020, GA No 952215), and the Collaboration Lab "AI in Construction" (AICO). It also benefited from the HMWK cluster projects "The Third Wave of AI" and "The Adaptive Mind" as well as the EU Project TANGO (Grant Agreement no. 101120763). The authors of the Eindhoven University of Technology received support from their Department of Mathematics and Computer Science and the Eindhoven Artificial Intelligence Systems Institute.

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

# Supplementary Materials

In the following, the interested reader can find details on evaluations and corresponding information.

## A  PIX2CODE DETAILS

### A.1  OBJECT REPRESENTATIONS.

Initially, the approach of [Chen et al., 2022] was developed to detect each object in an image and provide one class label to these. We modify this setting by training the model to predict multiple attributes per object. Hereby, each attribute is treated individually and is provided its own bounding box coordinates. The dynamic nature of the output of Pix2Seq allows to predict the same bounding box multiple times with another attribute class with just one prediction. In Pix2Code, to retrieve the symbolic representations $O$ from such a sequence of detected attributes, the attributes are combined based on their corresponding bounding boxes. For this, the values of the bounding box are compared with a tolerance of 7 to compensate for model errors, and similar bounding boxes get aggregated. Attribute labels associated with bounding boxes of one aggregation belong to the same object.

The object extractor of Pix2Code was independently trained for 50 epochs. We used the same hyperparameters as Chen et al. [2022] with ResNet50 as backbone but decreased the learning rate to 0.0003 and the learning rate of the backbone to 0.00003 and use a batch size of 8. We finetuned a pre-trained version of Pix2Seq[3] on respectively 2000 random images of Kandinsky Patterns and CLEVR images. Both image sets have two to ten objects in their scenes. We report the different mean Average Precision (AP) values for both data sets on five different seeds in Tab. 6 on RelKP and CLEVR Johnson et al. [2017] images.

The metric mAP calculates mean Average Precision values for ten different Intersection over Union (IoU) thresholds, from 0.50 to 0.95 in 0.05 steps. This rewards models with better localization of objects more. The metrics $AP_{50}$ and $AP_{75}$ give the average precision values of classifications with IoU values over 50% and 75%. The $AP_{75}$ values decrease only slightly in comparison to $AP_{50}$, which shows that there are few object detections where the IoU is under 75%. However, when comparing the $AP_{75}$ values to the AP value, one can see that the performance decreases quite a bit. This is, because the AP considers AP metrics with IoU bounds over 75% as well, where the model reaches its limits. The metrics $AP_S$, $AP_M$ and $AP_L$ measure the performance of the model on small, medium and large objects. The performance is best on large objects and decreases from medium to smaller ones which is typical for object detection models, as larger objects consist of more pixels and therefore provide more features based on which they can be classified.

Overall, Pix2Seq provides high AP values and is, therefore, a well-suited object extractor for our method.

Table 6: Average Precision of Pix2Seq approach with multiple attribute classes on RelKP and CLEVR images. Models have been fine-tuned on training examples with 5 different seeds. They have been evaluated on 750 test examples respectively.

| Dataset | AP | $AP_{50}$ | $AP_{75}$ | $AP_S$ | $AP_M$ | $AP_L$ |
|---------|-----|-----------|-----------|--------|--------|--------|
| RelKP | $89.7 \pm 0.45$ | $97.8 \pm 0.31$ | $96.4 \pm 0.49$ | $75.4 \pm 1.10$ | $91.8 \pm 1.08$ | $96.3 \pm 0.61$ |
| CLEVR | $96.5 \pm 0.26$ | $98.8 \pm 0.19$ | $98.7 \pm 0.16$ | $91.1 \pm 0.8$ | $96.7 \pm 0.24$ | $99.2 \pm 0.32$ |

### A.2  PROGRAM SYNTHESIS.

For the domain of visual concepts, the input of the program synthesis tasks has the type of a list of symbolic object representations, *i.e.* a integer list. Therefore, the domain specific language (DSL) for Pix2Code was created based on that from the list domain of Ellis et al. [2023] and adapted to construct concepts, *e.g.*, with primitives like *forall* and *count*. A list of all primitives used in our evaluations can be found in Tab. 7. The DSL has primitives like *fold* and *map* and logical primitives like *and* and *not*. Further, there are integer values which are needed for counting, but more importantly to encode the attribute values of the objects. In Tab. 8 we provide a mapping from integers to the attributes of RelKP and CURI objects is given.

For the main evaluations, Pix2Code's program synthesis component was trained with an enumeration timeout of 720s and 96 CPUs. We used the batching strategy "batch all unsolved" where the algorithm starts with all tasks and continues with the

---

[3]https://github.com/gaopengcuhk/Pretrained-Pix2Seq

Table 7: DSL used in Pix2Code experiments. t0 can be an arbitrary type.

| Primitive | Types | Description |
|---|---|---|
| true | bool | Boolean with positive value. |
| not | bool → bool | Boolean operator that negates a boolean. |
| and | bool → bool → bool | Boolean operator. If both inputs are true, the ouput is true. Otherwise it is false. |
| or | bool → bool → bool | Boolean operator. If at least one input is true, the output is true. Otherwise it is false. |
| eq? | int → int → bool | Compares two integer values. If they are equal the output is true. Otherwise it is false. |
| gt? | int → int → bool | Compares two integer values. If the first one is greather than the second one the output is true. Otherwise it is false. |
| find | t0 → list[t0] → int | Searches for the given element in the given list. Returns the index of the element in the list. Throws error if no element is found. |
| max | int → int → int | Given two integer values the function returns the higher value. |
| min | int → int → int | Given two integer values the function returns the lower value. |
| map | (t0 → t1) → list[t0] → list[t1] | Applies the input function to every element in the given input list of type t0. The output is a list of type t1. |
| index | int → list[t0] → t0 | Takes an integer and a list as input and outputs the element at the index defined by the input integer. |
| fold | list[t0] → t1 → (t0 → t1 → t1) → t1 | Inputs a list, a start value and a function. Applies the function to the start value and to the first element in the list and overwrites the start value by the result. Repeats this with every element in the list. |
| length | list[t0] → int | Returns the length of the given list. |
| if | bool → t0 → t0 → t0 | Inputs a boolean and two options. If the boolean is true, the first option is returned, else the second one. |
| + | int → int → int | Adds two integer values. |
| - | int → int → int | Subtracts two integer values. |
| empty | list(t0) | An empty list. |
| cons | t0 → list[t0] → list[t0] | Appends the given item to the start of the given list. |
| car | list[t0] → t0 | Returns the head of the given list. |
| cdr | list[t0] → list[t0] | Returns the tail of the given list. |
| empty? | list[t0] → bool | Returns true if the list is empty. Otherwise returns false. |
| forall | (t0 → bool) → list[t0] → bool | Takes a predicate function and a list and applies the function to all elements in the list. If for all the predicate is true, the function returns true. |
| exists | (t0 → bool) → list[t0] → bool | Takes a predicate function and a list and applies the function to all elements in the list. If for at least one element the predicate is true, the function returns true. |
| count | list[t0] → t0 → int | Takes a list and an element and counts how often the element appears in the list. |
| 0-9 | int | Integer values. |
| 10-14* | int | Integer values |

*Only used in revising confounded task experiment.

tasks that were not solved in the previous iterations. Further an $\lambda$-value of $1.5$, $\alpha$-value of $30$ and beam size of $5$ was used. The experiments were run for $15$ iterations.

## A.3  LEARNED LIBRARY PRIMITIVES

In Fig. 6 we provide a graph of all the abstracted primitives of a Pix2Code model trained on CURI iid. Let us highlight a few things here. We observe that one of the basic primitives is f0=($\lambda$ (x y) (forall ($\lambda$ (z) (eq? y (index x z))))). This represents a function that can be applied on a list of lists and checks for each element if the value at index x is equal to y. This is quite general and can be used to compare colors, shapes, etc.

In another primitive, f1=(f0 5), the primitive f0 gets extended so that the parameter x gets set by the value 5 (which represents the index for color). f1 is then integrated in other primitives, e.g. f8=(f1 6) and f18=(f1 0) where y is set respectively with the values 6 and 0 for the colors purple and gray. This means that f8 resembles the concept all objects are purple and f18 the concept all objects are gray.

Table 8: A mapping from integers to the attributes of RelKP and CURI objects.

| RelKP | | | CURI | | | |
|---|---|---|---|---|---|---|
| size | color | shape | size | color | shape | material |
| 0: small | 0: red | 0: triangle | 0: small | 0: gray | 0: cube | 0: rubber |
| 1: medium | 1: blue | 1: square | 1: large | 1: blue | 1: sphere | 1: metal |
| 2: big | 2: yellow | 2: circle | | 2: brown | 2: cylinder | |
| | | | | 3: yellow | | |
| | | | | 4: red | | |
| | | | | 5: green | | |
| | | | | 6: purple | | |

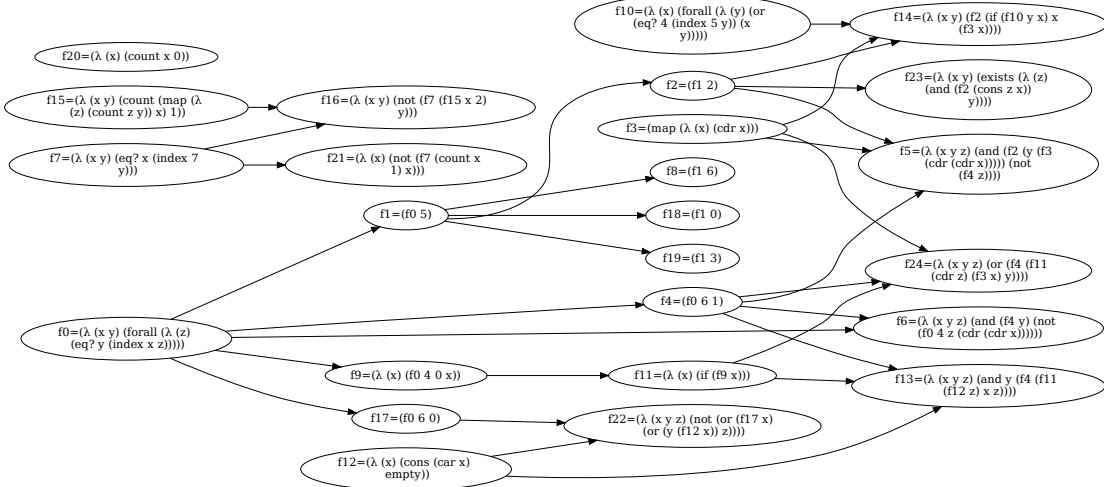

Figure 6: Graphical illustration of the library primitives abstracted by Pix2Code trained on CURI IID.

## A.4 CLASSIFYING AN IMAGE.

Given learned concept representations Pix2Code can identify if a specific concept is present in a novel image by first extracting the corresponding object representations and then testing a selected program of a concept on these. If the concept is present in the image the program will return True. Fig. 7 sketches this procedure.

## B CURI BASELINE MODEL

For comparing Pix2Code, we use the baseline model of Vedantam et al. [2021], which we refer to as CURI-B. For that we train the four different proposed architectures via the query loss of Vedantam et al. [2021](*i.e.* $\alpha = 0$) as we are investigating unsupervised concept learning settings in this work. We train for 1000 steps and use the same hyperparameters as used in the main evaluation of the authors. Based on Tab. 9 we select the best performing pooling approach for each dataset and split.

## C DATA USED FOR MAIN EXPERIMENTS

### C.1 RELKP

To investigate the relational concept learning abilities of Pix2Code, the data set RelKP was constructed, which includes 200 specific Kandinsky Patterns of varying complex concepts based on the number of objects, concept types, number of relations, and number of pairs. For each train concept, one support and one query set were created. The support and query

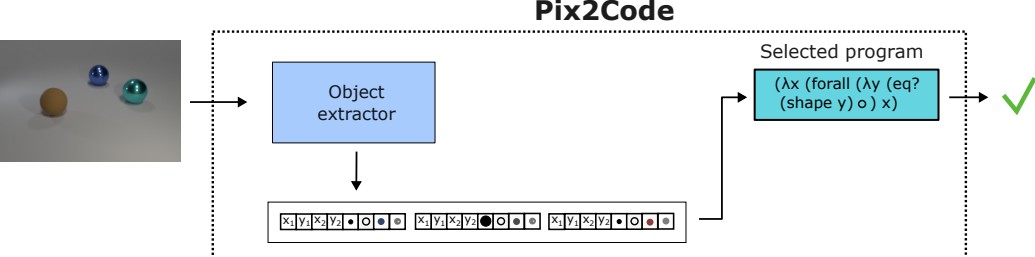

Figure 7: Pix2Code overview for the classification of an unseen image. The image gets processed by the object extractor resulting in a symbolic object representation. This is the input to the program used for classification which gives the final prediction.

Table 9: Mean accuracy (with std) for different datasets and splits reported individually for the model of [Vedantam et al., 2021] based on the different proposed pooling approaches.

|  | Concatenation | Global Avg. | Relation Net | Transformer |
|---|---|---|---|---|
| RelKP iid | $50.87 \pm 0.08$ | $\mathbf{59.69} \pm 0.83$ | $57.45 \pm 1.47$ | $52.76 \pm 0.24$ |
| CURI iid | $57.96 \pm 1.00$ | $65.57 \pm 0.91$ | $\mathbf{66.68} \pm 1.50$ | $62.24 \pm 1.54$ |
| Boolean | $56.63 \pm 0.80$ | $63.85 \pm 3.70$ | $\mathbf{67.86} \pm 1.21$ | $66.60 \pm 1.16$ |
| Counting | $54.48 \pm 1.90$ | $60.24 \pm 0.68$ | $58.87 \pm 2.16$ | $\mathbf{62.19} \pm 2.44$ |
| Extrinsic | $61.89 \pm 2.19$ | $67.81 \pm 5.48$ | $\mathbf{72.56} \pm 0.40$ | $70.21 \pm 2.84$ |
| Intrinsic | $53.01 \pm 0.13$ | $66.16 \pm 1.33$ | $\mathbf{67.85} \pm 2.50$ | $63.70 \pm 4.54$ |
| Binding(color) | $58.41 \pm 0.55$ | $65.20 \pm 3.05$ | $61.21 \pm 2.61$ | $\mathbf{69.89} \pm 1.54$ |
| Compositional | $59.16 \pm 1.95$ | $65.18 \pm 0.22$ | $\mathbf{67.63} \pm 0.53$ | $65.42 \pm 3.46$ |
| Complexity | $57.94 \pm 1.79$ | $64.04 \pm 1.31$ | $\mathbf{65.24} \pm 0.14$ | $62.27 \pm 1.74$ |
| Binding(shape) | $59.30 \pm 3.02$ | $57.24 \pm 4.67$ | $\mathbf{66.35} \pm 0.36$ | $63.51 \pm 3.45$ |

sets consist of 25 examples with five positive 20 negative image examples of the concept, respectively. The test concepts have one support set and eight query sets per concept, giving 40 positive and 160 negative examples for the query set. The generated patterns are inspired by the Kandinsky Patterns of Shindo et al. Shindo et al. [2023a] Example images can be seen in Fig. 8.

Table 10: Overview of relations that are used to create RelKP data set.

| Relation | Description |
|---|---|
| same_color | $\forall x, y : color(x) = color(y)$ |
| same_shape | $\forall x, y : shape(x) = shape(y)$ |
| same_size | $\forall x, y : size(x) = size(y)$ |
| one_red_triangle | $\exists x : color(x) = red \wedge shape(x) = triangle \wedge \forall y : x \neq y \wedge color(y) \neq red \wedge shape(y) \neq triangle$ |

There are two types of concepts, those where the relations refer to all objects in the image and those where the relations refer to only a pair of objects in the image. The relations include object concepts like "same shape" and "one object is a red triangle". The used relations are listed in Tab. 10. The relations can be combined with and and or and not can be applied to relations.

In RelKP the smallest number of objects is two and the maximum number is six. For concepts where a relation refers to all objects all objects in an image have to indicate the specific concept. For concepts where the relations only refer to a pair, this means that among all objects there should exist at least one pair for which the objects have the same shape and the same color. More complex patterns of this type can have relations for the remaining distinct pairs of objects in the image as well. An example of a clause describing such a pattern is given in equation Eq. 8, *i.e.*, there is a pair of objects that has the same shape and the same color, there is another distinct pair that also has the same shape and the same color and there is a third pair that does not has the same shape or it does not have the same color.

Two objects that have the same color

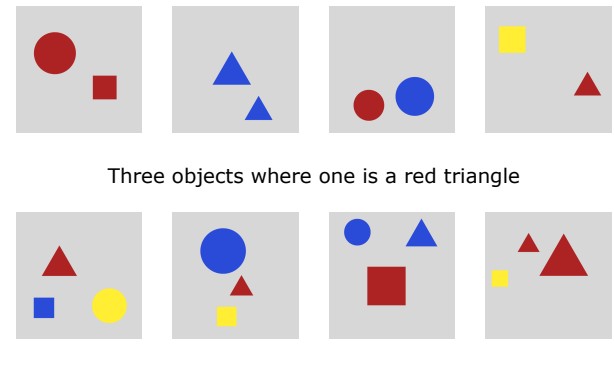

Three objects where one is a red triangle

One pair with same shape and one pair with not the same shape

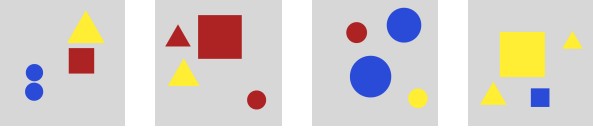

Figure 8: Three example concepts of RelKP. The two left images depict positive examples of the concept and the two right images depict negative ones.

$$\exists x_1 \exists x_2 \exists y_1 \exists y_2 \exists z_1 \exists z_2$$
$$((x_1 \neq x_2) \wedge (x_1 \neq y_1) \wedge (x_1 \neq y_2) \wedge (x_1 \neq z_1) \wedge (x_1 \neq z_2)$$
$$\wedge (x_2 \neq y_1) \wedge (x_2 \neq y_2) \wedge (x_2 \neq z_1) \wedge (x_2 \neq z_2)$$
$$\wedge (y_1 \neq y_2) \wedge (y_1 \neq z_1) \wedge (y_1 \neq z_2)$$
$$\wedge (y_2 \neq z_1) \wedge (y_2 \neq z_2) \wedge (z_1 \neq z_2)$$
$$\wedge (same\_shape(x_1, x_2) \wedge same\_color(x_1, x_2))$$
$$\wedge (same\_shape(y_1, y_2) \wedge \neg same\_color(y_1, y_2))$$
$$\wedge (\neg same\_shape(z_1, z_2) \vee \neg same\_color(z_1, z_2)))$$

(8)

## C.2 CURI

The **CURI** dataset [Vedantam et al., 2021] is based on CLEVR images [Johnson et al., 2017], which depict 3D objects that possess the attributes *color*, *shape*, *size* and *material* (*cf.* Fig. 1 for example images). The dataset has a total number of 14 929 abstract concepts. For each concept, the dataset contains at least one *episode*, which consists of a support and a query set of images, each with five positive and 20 negative image examples. Overall, the data set is designed to test for compositional generalization and thus contains eight different concept splits that are based on specific properties that occur only in the test set.

The "counting" split tests for counting generalization via 47 novel combinations of property-count concepts in its test set. There are the intrinsic and extrinsic property splits, where in the training set concepts like "green" and "metal" or "red" and position "1" (on the x or y axis) do not occur together. For the boolean split there occur some combinations of properties and logical operators only in the test split, *e.g.*, "green" and "or". Further, the binding splits have some object attributes only in the test concepts, *i.e.*, the shape cylinder occurs only in test concepts for Binding(shape) and the colors purple, cyan and yellow occur only in the test concepts for Binding(color). For the counting split, there is a selection of 47 concepts that are counting-based in the test set, but still some other counting concepts in the train set as well. The complexity split takes only concepts that are shorter than 10 tokens (*i.e.*, that are less complex) for training and the longer ones for testing. We refer to the original work [Vedantam et al., 2021] for further details.

# D  DETAILS ON EXPERIMENTAL EVALUATIONS

In the following, we provide additional experimental details, but importantly, also ablation evaluations where both CURI-B and Pix2Code are provided with ground-truth object information input rather than the raw images. Vedantam et al. [2021] refer to this type of input as *schema* input.

All experiments were performed using the following hardware: CPU: AMD EPYC 7742 64- Core Processor, RAM: 2064 GB, GPU: NVIDIA A100-SXM4-40GB GPU with 40 GB of RAM.

## D.1  CHOICE OF CURI SUPPORT SETS FOR PIX2CODE

The CURI data set is constructed in a way that a model can predict labels for a query set based on support examples. Therefore, for one concept in CURI there exists often multiple support sets with respective query sets. Our Method Pix2Code works different in a sense that it retrieved a program based on a support set of examples and based on that is able to classify arbitrary query examples, *i.e.*, executing the program on them. To evaluate Pix2Code on the CURI dataset we therefore chose to consider just one support set per concept to reduce the number of programs that need to be enumerated per concept. In Tab. 11, we analyze if this changes the performance of Pix2Code whereby we show that this doesn't affect the model's performance notably. In our evaluations of Pix2Code we therefore consider only one support set per concept.

Table 11: Comparison of one support set per task and different support sets per task. Mean Acc@all of Pix2Code on test tasks of CURI splits with schema inputs are reported.

| CURI Splits 100 | Pix2Code (same support) | Pix2Code (diff support) |
|---|---|---|
| Boolean | 80.58 | 80.39 |
| Counting | 58.24 | 57.77 |
| Extrinsic | 77.51 | 78.03 |
| Intrinsic | 89.25 | 89.93 |
| Binding(color) | 80.89 | 81.04 |
| Compositional | 77.45 | 77.77 |
| Binding(shape) | 78.35 | 77.58 |
| Complexity | 73.42 | 73.52 |

## D.2  LEARNING VISUAL CONCEPTS.

Tab. 12 presents the results of an ablation study where we provide ground truth symbolic representations of the objects in each image for the RelKP iid and CURI iid split, rather than the representations of Pix2Code's object extractor or CURI-B's image encoder. We observe that Pix2Code represents a competitive approach over CURI-B particularly when considering the accuracy of the solved tasks.

Tab. 13 (top) presents how many tasks Pix2Code has solved per dataset. Leading to an average of $93\%$ for RelKP iid and $68.86\%$ for CURI iid. For schema inputs Tab. 16 (top) the average of solved tasks for RelKP is $93.67\%$ and for CURI $72.42\%$, which is slightly higher.

Table 12: Mean test accuracy on Kandinsky and CURI concepts with iid train test splits and *schema* input.

| | CURI-B | Pix2Code (Acc@all) | Pix2Code (Acc@solved) |
|---|---|---|---|
| RelKP | **91.36** $\pm0.59$ | 91.01$\pm0.90$ | 93.95 $\pm0.51$ |
| CURI (iid Split) | 73.73 $\pm0.31$ | **74.16** $\pm1.18$ | 83.31 $\pm1.65$ |

Table 13: Number of solved CURI concepts for *image* input with Pix2Code over the three seeds.

| Datasets | Seed 0 | Seed 1 | Seed 2 | Avg. | Total Tasks |
|---|---|---|---|---|---|
| RelKP iid | 91 | 94 | 95 | 93 | 100 |
| CURI iid | 7005 | 4936 | 5389 | 5777 | 8389 |
| Boolean | 1665 | 2002 | 1776 | 1814 | 2565 |
| Counting | 9 | 20 | 19 | 16 | 47 |
| Extrinsic | 540 | 501 | 632 | 558 | 750 |
| Intrinsic | 258 | 272 | 223 | 251 | 283 |
| Binding(color) | 1927 | 1940 | 2211 | 2026 | 2590 |
| Compositional | 1699 | 1597 | 1605 | 1634 | 2402 |
| Complexity | 6739 | 6811 | 6914 | 6821 | 8363 |
| Binding(shape) | 1002 | 789 | 1165 | 985 | 1484 |

## D.3 TIME COSTS OF PIX2CODE

In Tab. 14 we provide the mean durations (in sec.) for the training of CURI-B and Pix2Code (and its sub-modules) on the iid data set of CURI over three seeds.

Table 14: Training times of CURI and Pix2Code.

| | CURI-B | Pix2Code | *Object Extractor* | *Program Synthesis* |
|---|---|---|---|---|
| Duration | $1094.3 \pm 122$ s | $48575.7 \pm 2338$s | $17247.3 \pm 583$s | $31328.3 \pm 2404$s |

The training of CURI-B for 1000 steps takes, on average, 1094 seconds (ca. 18 minutes). Pix2Code was trained for 15 iterations which takes on average $48,575.5$ seconds (ca. 13, 5h). Indeed, this is a substantially longer training time. However, we consider this to be a trade-off for the benefits of improved generalisability, interpretability, and revisability.

## D.4 GENERALIZING TO NOVEL COMBINATIONS OF KNOWN VISUAL CONCEPTS.

In Tab. 15 we provide the ablation results when both models are trained on schema input. We observe the same trend as in the evaluations of the main text. Tab. 13 (bottom) presents how many tasks Pix2Code has solved per CURI split. Leading to a median of $72.56\%$ over all splits. For schema inputs Tab. 16 (bottom) the median is $74.95\%$.

Table 15: Mean accuracy (with std) for meta-test tasks of CURI splits reported individually and as the median (with median absolute deviation) over all splits. Hereby the models were provided with *schema* inputs, rather than images.

| CURI (Comp. Splits) | CURI-B | Pix2Code (Acc@all) | Pix2Code (Acc@solved) |
|---|---|---|---|
| Boolean | 75.69 $\pm 0.41$ | **80.46** $\pm 1.28$ | 91.07 $\pm 2.39$ |
| Counting | **70.56** $\pm 0.44$ | 58.34 $\pm 0.45$ | 69.16 $\pm 2.81$ |
| Extrinsic | 76.97 $\pm 0.18$ | **78.67** $\pm 1.82$ | 89.68 $\pm 1.93$ |
| Intrinsic | 78.18 $\pm 1.61$ | **87.47** $\pm 3.34$ | 92.64 $\pm 0.98$ |
| Binding(color) | 78.37 $\pm 0.61$ | **80.61** $\pm 2.27$ | 87.62 $\pm 2.49$ |
| Compositional | 74.12 $\pm 0.91$ | **77.19** $\pm 0.52$ | 87.21 $\pm 0.68$ |
| Binding(shape) | 72.60 $\pm 0.82$ | **78.75** $\pm 1.68$ | 88.33 $\pm 2.26$ |
| Complexity | **75.07** $\pm 0.43$ | 74.21 $\pm 0.56$ | 78.43 $\pm 0.56$ |
| Mdn. | 75.38 $\pm 2.19$ | **78.71** $\pm 1.82$ | 87.98 $\pm 2.40$ |

Table 16: Number of solved CURI concepts for *schema* input with Pix2Code over the three seeds.

| Datasets | Seed 0 | Seed 1 | Seed 2 | Avg. | Total Tasks |
|---|---|---|---|---|---|
| RelKP iid | 92 | 95 | 94 | 94 | 100 |
| CURI iid | 6875 | 5482 | 5869 | 6075 | 8389 |
| Boolean | 1769 | 2119 | 1840 | 1909 | 2565 |
| Counting | 23 | 26 | 20 | 23 | 47 |
| Extrinsic | 498 | 498 | 634 | 543 | 750 |
| Intrinsic | 263 | 274 | 211 | 249 | 283 |
| Binding(color) | 1963 | 2070 | 2300 | 2111 | 2590 |
| Compositional | 1817 | 1754 | 1696 | 1756 | 2402 |
| Complexity | 7085 | 7060 | 7132 | 7092 | 8363 |
| Binding(shape) | 1130 | 963 | 1267 | 1120 | 1484 |

## D.5 GENERALIZING TO VARIABLE NUMBER OF OBJECTS.

For investigating entity generalization in the context of visual concept learning we created images with the CLEVR-Hans repository [Stammer et al., 2021] for generating the CURI variations AllCubes-N and AllMetalOneGray-N. In AllMetalOneGray-N positive images all contain metal objects and at least one gray object. Negative images have a rubber object and others are metal. Examples of the datasets are depicted in Fig. 9.

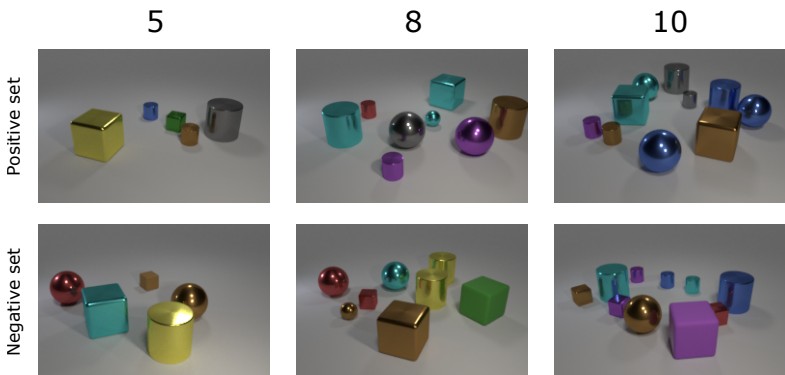

Figure 9: Examples of created test examples for AllMetalOneGray-N. Positive images have all metal objects and at least one gray one. Negative images have one rubber object

For the support sets of AllCubes-N and AllMetalOneGray-N, we used one original, randomly sampled support set of the concepts "all objects are cubes" and "all objects are metal and there exists a gray object" from the CURI data set. For the query sets 100 positive and 100 negative examples were created and grouped into 25 examples per query set.

For the evaluations CURI-B needed to be retrained on the iid train split with the hyperparameter max objects set to 10, leading to CURI-B-10. The best performing model was the one with transformer pooling, its test results on the CURI iid split are reported in Tab. 17. For Pix2Code, the original trained models from the iid split were used to query a program for the support sets of the concepts and classify the query examples. Both models achieve comparable results on the original data set, however, the evaluations of (Q3) in (Tab. 3) show that in terms of entity generalization Pix2Code largely outperforms CURI-B-10.

## D.6 INTERPRETING PROGRAMS.

For providing the natural language explanations of Tab. 4 we used gpt-4-turbo. An exemplary prompt for "all objects are cyan" is shown in Listing 1. Note that the prompt uses the raw program output whereas in the Tab. 4 and Tab. 18 the programs where parsed to a more readable form and integer values were substituted with their semantic meaning.

Table 17: Mean test accuracy on CURI concepts with iid train test splits and *schema* input where CURI-B is modified to process up to ten objects (CURI-B-10).

|  | CURI-B-10 | Pix2Code (Acc@all) | Pix2Code (Acc@solved) |
|---|---|---|---|
| CURI (iid Split) | 74.5 $\pm_{1.55}$ | 74.16 $\pm_{1.18}$ | 83.31 $\pm_{1.65}$ |

Further, we evaluated the language model Gemini[4] for comparison, the results are given in Tab. 18.

Table 18: Examples of CURI concepts from with Pix2Code programs and natural language translation from an LLM, *i.e.*, gemini-pro-dev-api. All programs achieve 100% accuracy on the CURI test scenes.

| Original Concept | Retrieved Program | Natural Language Translation (Gemini) |
|---|---|---|
| **All** objects are **cyan** | (λ (x) ((λ (y z) (**forall** (λ (u) (eq? (index y u) z)))) **color cyan** x)) | "All objects have the color cyan." |
| **All** objects are **cubes** | (λ (x) ((λ (y z) (**forall** (λ (u) (eq? (index y u) z)))) **shape cube** x)) | "All objects have the shape cube." |
| **All** objects are **purple and all** objects are **spheres** | (λ (x) (**and** ((λ (y z) (**forall** (λ (u) (eq? (index y u) z)))) **shape sphere** x) ((λ (v w) (**forall** (λ (a) (eq? (index v a) w)))) **color purple** x))) | "All objects are spheres and purple." |
| **All** objects are **brown or all** objects are **cubes** | (λ (x) (**or** ((λ (y z) (**forall** (λ (u) (eq? (index y u) z)))) **shape cube** x) ((λ (v w) (**forall** (λ (a) (eq? (index v a) w)))) **color brown** x))) | "All objects are either cubes and made of rubber or are brown." |
| **All** objects are **small and there exists** a **purple** object | (λ (x) (**and** ((λ (y z) (**forall** (λ (u) (eq? (index y u) z)))) **size small** x) (**exists** (λ (v) ((λ (w a b) (eq? (index b w) a)) v **purple color**)) x))) | "All objects are small and there exists a green object." |
| There are **three gray** objects | (λ (x) (**eq?** ((λ (y) (**count** (map (λ (z) ((λ (u v) (index u v)) **color** z)) y))) x **gray**) **3**)) | "All objects have size large." |
| There exists an arbitrary object and **there exist three** other objects that are **blue** | (λ (x) (**gt?** ((λ (y) (**count** (map (λ (z) ((λ (u v) (index u v)) **color** z)) y))) x **blue**) **2**)) | "There are more than 2 objects with size large." |

[4]https://blog.google/technology/ai/google-gemini-ai

Listing 1: Example prompt for LLMs.

There is a list of integer lists that represent objects from an image.
Each object is encoded by four values for the bounding box of the object,
then one value for the size, one value for the color, one for the shape
and one for the material. This means an object is encoded by a list of 8 values:
[x_min, y_min, x_max, y_max, size, color, shape, material].

The values for size (index 4):
0: small
1: large

The values for color (index 5):
0: gray
1: blue
2: brown
3: yellow
4: red
5: green
6: purple
7: cyan

The values for shape (index 6):
0: cube
1: sphere
2: cylinder

The values for material (index 7):
0: rubber
1: metal

In the following there is a lambda calculus program that processes a list of objects
and classifies them based on a rule. The rule determines whether the image belongs
to a pattern or not (True or False).

Please give description of the pattern that is detected by the program in one
sentence.

Program:
(lambda (#(#(lambda (lambda (forall (lambda (eq? (index $2 $0) $1))))) 6 1) $0))

Explanation:
All objects have the shape sphere.

Program:
(lambda (#(#(#(lambda (lambda (forall (lambda (eq? (index $2 $0) $1))))) 5) 2) $0))

Explanation:
All objects have the color brown.

Program:
(lambda (#(#(lambda (lambda (forall (lambda (eq? (index $2 $0) \$1))))) 5) 7 $0))

Explanation:

## D.7 REVISE CONFOUNDERS.

For the evaluations of confounding in concept learning, we propose **CURI-Hans**. It consists of original CURI concepts listed in Tab. 19 and a confounded test task. This test task is confounded by "all objects are cyan", which is added to the support set of test task, *i.e.* each object gets the color *cyan*. The query set stays unconfounded, *i.e.*, every object can have any color from the set of existing colors of the original CLEVR setting.

Table 19: Subset of CURI concepts for confounded experiment. For each concept, one episode was selected and the test task has been confounded so that in the support set the positive samples had **all cyan objects.**

| Split | Concept | Description |
|-------|---------|-------------|
| Train | 6746 | All objects are blue |
| Train | 6001 | All objects are spheres |
| Train | 7666 | There exists a metal object and its x-location is greater than 1 |
| Train | 4399 | There exists a sphere and there exists an object with y-location equal to 7 |
| Train | 9659 | There exists a metal object and there exists another object which has the y-location 6 |
| Train | 14275 | All objects are brown and all objects are cylinders |
| Train | 13983 | All objects are red and there exists a cube |
| Train | 2524 | There exists a yellow object and all objects are rubber |
| Test | 5327 | There exists a cube and all objects are **(cyan and)** metal |

For revising the program synthesis component of Pix2Code, we remove program primitives from $L$, as well as collected programs of the training tasks that include the removed program primitives (as the code model $q_\psi$ is trained on them). To do this more easily, we change the object representations so that each object property has its own integer values, leading to Tab. 20 (in comparison to Tab. 8).

Table 20: A mapping from integers to the attributes of CURI-EG objects.

| CURI size | color | shape | material |
|-----------|-------|-------|----------|
| 0: small | 2: gray | 9: cube | 12: rubber |
| 1: large | 3: blue | 10: sphere | 13: metal |
| | 4: brown | 11: cylinder | |
| | 5: yellow | | |
| | 6: red | | |
| | 7: green | | |
| | 8: purple | | |
| | 9: cyan | | |

To remove the confounder *color cyan*, we can therefore remove the primitive *9* as well as the color index *5* (because color is at index 5 in the object representations). We finally finetune the code model on the modified library and reevaluate on the unconfounded query set.

## D.8 REVISE COUNTING.

To revise Pix2Code for the counting split of CURI, we add four primitives to the library $L$. These primitives are each designed to count the number of occurrences for a given attribute (*i.e. size, color, shape* and *material*) in a object representation list. The primitives are the following:

`(λ (x)(λ (y)(count (map (λ (z) (index 4 z))y)x)))` to count the times `y` occurs as size.

`(λ (x)(λ (y)(count (map (λ (z) (index 5 z))y)x)))` to count the times `y` occurs as color.

`(λ (x)(λ (y)(count (map (λ (z) (index 6 z))y)x)))` to count the times `y` occurs as shape.

`(λ (x)(λ (y)(count (map (λ (z) (index 7 z))y)x)))` to count the times `y` occurs as material.

The four primitives are added with prior probability of $-0.3$ to $L$. After that, the module is trained for another iteration to update the code model and Pix2Code is evaluated again on the test tasks of the counting split, achieving a much higher accuracy.

### D.9 EXTENDING PIX2CODE TO NATURAL IMAGES.

For evaluating Pix2Code on real-world concepts we created 7 abstract concepts based on the MS COCO dataset [Lin et al., 2014], an example concept is given in Fig. 5. Since we are only investigating the potential of applying Pix2Code to real-world scenarios, we consider a small set of training tasks and do not investigate the generalization to test tasks here. The training tasks have 25 support images based on which programs are retrieved and 100 images for testing the programs. In Tab. 5 the 5 COCO concepts for which Pix2Code synthesized programs are presented. The two concepts for which no program was retrieved are "There exist two dogs" and "There exists a book or a teddy bear".

In the context of integrating Pix2Code to more realistic image settings let us discuss potential bottlenecks and updates to mitigate these. Specifically, more realistic image settings can lead to an increased number and complexity of the object token sequences of the object extractor. However, this should not represent a bottleneck within the object extraction module of Pix2Code as object extractors like Pix2Seq can handle such settings as was illustrated in the original work [Chen et al., 2022].

In comparison, the program synthesis module could contain the following possible limitations. First off, given a large symbolic representation space, the code model can experience issues processing the entire symbolic input. However, approaches exist to mitigate this, e.g., an attention-based module could be used. Second, a large symbolic representation space may require more base program primitives (*e.g.*, more integer values), which can lead to an increased search time due to the larger search space. This may limit the applicability in time-sensitive settings. Concerning this, Ellis et al. [2023] suggests increasing the number of CPUs, which allows for parallelizing the searches. If time is not of the essence one can increase the search timeout parameters. Another possible measure to mitigate long search times is to incorporate a form of pre-filtering of objects and their attributes, thereby reducing the search space. In the case of our evaluations in (Q6), we used the same hyperparameters as for the other evaluations and used 16 CPUs to obtain the results, however, in the case of a higher number of real-world training tasks, it still needs to be investigated whether the search time and number of CPUs need to be increased.

We note that the current architecture of Pix2Code does not explicitly handle the object extractor's noise. If the noise is too high, it can happen that Pix2Code does not find a suitable program as in the case of the two COCO concepts. We propose to integrate object representation uncertainty in future work to apply Pix2Code to real-world settings in a robust way.