# OpenReview forum: "Pix2Code: Learning to Compose Neural Visual Concepts as Programs"
_auai.org/UAI/2024/Conference — UAI 2024 oral_

### Official Review · Reviewer_TaT1 · 2024-03-02

**Q2-1 Originality-Novelty:** 3
**Q2-2 Correctness-Technical Quality:** 3
**Q2-5 Clarity Of Writing:** 3

**Q1 Summary And Contributions:**

This paper proposed Pix2Code which aims to improve visual concept learning capability via learning to transform the latent invariant visual concepts into lambda-calculus programs. The paper evaluates Pix2Code on its prediction accuracy, its generalization towards novel combinations of latent concepts, and a larger number of objects in the images. Additionally, the paper qualitatively studies the interpretability of the synthesized programs of Pix2Code.

**Q2-3 Extent To Which Claims Are Supported By Evidence:**

3: Good: the main claims are supported by convincing evidence (in the form of adequate experimental evaluation, proofs, (pseudo-)code, references, assumptions).

**Q2-4 Reproducibility:**

4: Excellent: key resources (e.g. proofs, code, data) are available and key details (e.g. proof sketches, experimental setup) are comprehensively described for competent researchers to confidently and easily reproduce the main results.

**Q3 Main Strengths:**

* The idea of using neural program synthesis to enhance visual concept learning is novel
* Comprehensive evaluation, and the technique obtained good performance improvement over baseline
* The paper is overall well-written and structured, with extensive details

**Q4 Main Weakness:**

* Practicality of the approach seems to be limited

**Q5 Detailed Comments To The Authors:**

* The evaluated scenario seems to be a bit simple, where the objects have only a few attributes and concepts. What would be the possible bottleneck for the approach if it is applied to a more complex environment with more concepts and diverse objects?
* Why is the comparison with prior work only made against the purely neural model of Vedantam et al. [2021], but omitted the others, e.g., Kim et al., 2018, Stabinger et al., 2021?
* In the introduction, can you be more specific on what current ML approaches have achieved, their remaining issues, and list the contribution more clearly?

**Q9 Complying With Reviewing Instructions:**

Yes

---

> ### Author Rebuttal · Authors · 2024-04-05
>
> **1. Discuss bottleneck for more complex input.**
>
> We agree that this is valuable information. We kindly refer here to the general remark on this as well as our novel evaluations on real-world COCO images. We have added the corresponding discussions to our work.
>
> **2. Comparison to other related works.**
>
> Indeed, Kim et al. propose a method for few-shot learning scenarios as well, however their proposed Siamese architecture is designed for predicting the presence of concepts it has been trained on. Thus, it generalizes to unseen image samples of a concept, but not to unseen concepts which is one of the important aspects of our evaluations.
>
> Stabinger et al. (2021) represents a survey of differnt deep learning approaches on visual reasoning tasks but do not propose a specific architecture. They do postulate that Relation Network (RN) based approaches work on average best for the domain of relational reasoning. However, the baseline architecture, CURI-B, includes RNs as one of the pooling options (c.f. Table 8 in appendix). Therefore, we consider CURI-B to posit a valuable representative of purely neural models that can handle the visual concept learning setting of our evaluations.
>
>
> **3. Specifics in introduction concerning related works.**
>
> We agree. Based on the details of the previous answer we have added a more detailed discussion of the related works mentioned in our introduction. Moreover, we have added references and discussions of Webb et al. [1], Kerg et al. [2] and Vaishnav et al. [3].
>
> [1] Webb, Taylor, Shanka Subhra Mondal, and Jonathan D. Cohen. "Systematic visual reasoning through object-centric relational abstraction." NeurIPs (2024).
>
> [2] Kerg, Giancarlo, et al. "On neural architecture inductive biases for relational tasks." arXiv (2022).
>
> [3] Vaishnav, Mohit, and Thomas Serre. "Gamr: A guided attention model for (visual) reasoning." ICLR (2023).

---

### Official Review · Reviewer_a1Cx · 2024-03-21

**Q2-1 Originality-Novelty:** 3
**Q2-2 Correctness-Technical Quality:** 3
**Q2-5 Clarity Of Writing:** 4

**Q1 Summary And Contributions:**

This paper proposes Pix2Code, which aims at representing visual concepts in a generalizable, interpretable, and revisable manner. Towards this goal, it learns a library of programmes that operate on detected objects from input images. The aim of the programmes is to tell apart a positive and a negative set corresponding to a visual concept. The object detector, library of operators, and the programme generator are jointly optimized on synthetic image data. Empirical study on synthetic dataset supported the claims in limited domains.

**Q2-3 Extent To Which Claims Are Supported By Evidence:**

3: Good: the main claims are supported by convincing evidence (in the form of adequate experimental evaluation, proofs, (pseudo-)code, references, assumptions).

**Q2-4 Reproducibility:**

3: Good: key resources (e.g. proofs, code, data) are available and key details (e.g. proofs, experimental setup) are sufficiently well-described for competent researchers to confidently reproduce the main results.

**Q3 Main Strengths:**

1. The claim of the paper on the proposed Pix2Code method on its interpretability, generalizability, and revisability, are well supported by experiments on synthetic datasets.
2. I think, in principle, the proposed method can be applied to more complicated visual scenes, given that the object detector has sufficient capabilities. However, it would be better if this point had been empirically shown in the paper.
3. The paper is well written and easy to follow.

**Q4 Main Weakness:**

1. The proposed method is only evaluated on datasets with synthetic images and is largely limited to the location, shape, and colour of the objects. Its effectiveness is also only empirically proved therein. Therefore, it is unclear how Pix2Code performs in real-world images which are much more diverse and noisier.
2. The paper did not mention what resource required to train the models and how does this resource requirement change with respect to the complexity of the input images (e.g., how many objects and number of attributes).

**Q5 Detailed Comments To The Authors:**

I would like the authors to discuss more the complexity of the method in terms of its input. This provides useful clues on how well it can be used in more complicated real-world cases.

**Q9 Complying With Reviewing Instructions:**

Yes

---

> ### Author Rebuttal · Authors · 2024-04-05
>
> **1. Real-World images.**
>
> Thanks for pointing this out. We have added a evaluation on the COCO data set which can be found in the general comment above. We have added these to our manuscript.
>
> However, we do note that Pix2Code is not limited to the attributes (location, shape, color etc.) that were used for RelKP and CURI. The method is agnostic to the attributes that are provided by the object extractor and DSL such that more and other attributes can be used.
>
> **2. Resource requirement changes with more complex input.**
>
> We kindly refer to the general remark on this above. We have added the corresponding discussions to our work.
>
> **3. Used resources.**
>
> Thanks for pointing this out, we agree that this is a relevant information which we have now added to the appendix.
> The experiments were run on a machine with following hardware: CPU: AMD EPYC 7742 64- Core Processor, RAM: 2064 GB, GPU: NVIDIA A100-SXM4-40GB GPU with 40 GB of RAM.
>
> We further conducted evaluations on computation times and present in the following the mean durations (in sec.) for the training of CURI-B and Pix2Code (and its sub-modules) on the iid data set of CURI:
>
>
> |          | CURI-B          | Pix2Code | *Object Extractor* | *Program Synthesis* |
> | -------- | --------         | -------- | -------- | -------- |
> | Duration | 1094.3+-122s     | 48575.7+-2338s     | 17247.3+-583s | 31328.3+-2404s |
>
> The training of CURI-B for 1000 steps takes on average 1094 seconds (\~18 minutes). Pix2Code was trained for 15 iterations which takes on average 48,575.5 seconds (\~13,5h). This is a substantantially longer training time, however, we consider this to be a trade-off for the benefits of improved generalisability, interpretability and revisability.
>
> We added the computing time evaluation a discussion of it to the Appendix.

---

### Official Review · Reviewer_sdtM · 2024-03-23

**Q2-1 Originality-Novelty:** 3
**Q2-2 Correctness-Technical Quality:** 3
**Q2-5 Clarity Of Writing:** 4

**Q1 Summary And Contributions:**

A neurosymbolic framework called Pix2Code is proposed for classifying images based on attributes of the objects present in them. Pix2Code chains two components:
(1) an object extraction system, which represents each image as a set of objects with interpretable discrete-valued attributes;
(2) a program synthesis system, which takes as input the object-based representations of "positive" and "negative" examples for a compositional concept and finds the simplest logical formula that explains the split.
This is tested mainly on the CURI dataset, which uses CLEVR images. Various expierments show that classifiers built using Pix2Code classify and generalize better than end-to-end neural ones, and the symbolic formulae produced by the system can be converted into interpretable descriptions.

**Q2-3 Extent To Which Claims Are Supported By Evidence:**

3: Good: the main claims are supported by convincing evidence (in the form of adequate experimental evaluation, proofs, (pseudo-)code, references, assumptions).

**Q2-4 Reproducibility:**

4: Excellent: key resources (e.g. proofs, code, data) are available and key details (e.g. proof sketches, experimental setup) are comprehensively described for competent researchers to confidently and easily reproduce the main results.

**Q3 Main Strengths:**

- Overall, the paper studies a very relevant problem for the community, namely, the combination of rule-based systems, which are interpretable and generalizable, with neural components. The neural-symbolic interface is featured in (at least) three ways in the paper:
  - Neural feature extraction to produce the image representations that are input to the programs.
  - Guidance of the program search by a neural model, using the DreamCoder framework.
  - Though not a primary component of the system, use of a LLM to convert the programs to expressions in language.
- The paper is very well written (in terms of organization, clarity, references to related work). It gives a good introduction to the problem for anyone who is modestly familiar with object-centric representations, program synthesis, or both.
- Very thorough experimental analysis; the choice of problems is appropriate for verifying the stated hypotheses.

**Q4 Main Weakness:**

- Although the combination of components is new, the technical novelty is somewhat limited. All the main components (DreamCoder, feature extraction) are used out of the box.
  - In my opinion, this is outweighed by the novel way to combine these components and the good analysis of the reliance on the concept library, etc.
- Uncertainty is not modeled by the system:
  - Less significantly for the main ideas of the paper, the feature extraction is deterministic (so ambiguous images are not handled in the proper Bayesian / empirical risk minimization way).
  - More importantly, uncertainty over the programs is not modeled. Modeling the full Bayesian posterior / multiple candidates over hypotheses that explain the data equally well will (provably) improve generalization. (Are there problems in the dataset where multiple explanations, of similar length, yield optimal classification?)
- Problems considered are also a little simple, although I understand that the authors needed to isolate the difficulties of object representation learning from the problem of inducing compositional concepts.
- The CURI-B baseline may not be optimally illustrating the benefits of the neurosymbolic approach. If I understand correctly, CURI-B is using image inputs with a (still differentiable) object bottleneck, but does not pass through the pretrained discrete representation used as input to programs in Pix2Code.
  - The comparison does not show if the neurosymbolic approach is working well because the good object representations are used or because symbolic expressions, and not neural networks, are used as the classifiers.
  - Possible experiment: what happens if train a neural model is trained to take the same discrete object representations as input?

**Q5 Detailed Comments To The Authors:**

- Could you please comment on the following?
  - How could the classifier guide the learning of the object encoder (in the setting of unsupervised object+attribute extraction)?
  - How can extensions of Pix2Code be useful on harder image tasks, where object-centric image representations (e.g., ones derived from slot attention) are less interpretable and may contain some real-valued attributes?
- Can you give some examples of the abstracted primitives found by DreamCoder?

**Q9 Complying With Reviewing Instructions:**

Yes

---

> ### Author Rebuttal · Authors · 2024-04-05
>
> **1. Feature extraction deterministic.**
>
> We agree that it would be interesting to investigate how the framework can be extended to better handle uncertainty in the feature extraction for future work. Specifically, we consider integrating ideas of probabilistic programming [1] to be a noteworthy starting point for future improvements.
>
> **2. Modeling uncertainty over programs**
>
> Thanks for pointing this out. We see that the description of the enumarative search over programs in our methods section is slightly missleading. Actually the code model models the posterior over programs given the task examples by giving a distribution over the program primitives. The posterior thereby results from the combined probabilities of the primitives. During the enumerative search the probabilities of the programs under $q_{\phi}$ are memorized and the output of the search is the program with the highest probability. This stands in contrast to solely sampling a program from $q_{\phi}$.
>
> We acknowledge that this part was not specific enough in our method section and we have updated it. Specifically, we have made clear that the retrieved program is the program with the highest posterior found during enumeration.
>
> **3. Ablation.**
>
> Yes, we fully agree. In fact, an ablation study where both models received ground truth object representations as input (schema) can be found in Table 13 in the appendix. We observe that Pix2Code overall performs better than CURI-B even when CURI-B is provided GT object representations. Moreover, Pix2Code only loses ~2.14% median accuracy when provided the imperfect, predicted object representations in Table 2. This indicates that the symbolic expressions are indeed useful for generalization and not *just* the good object representations.
>
> **4. Guiding object extractor through reasoning.**
>
> Indeed this is an interesting question. One interesting option would be to integrate differentiable programming [2] or even differentiable, probabilistic programming approaches [1,3] into program synthesis. Particularly, the later would not just allow to train the neural, object extractor via the reasoning task, but even to handle uncertainty in the visual perception module.
>
> A second possible direction could be to synthesize programs for multiple subsets of a task to handle possibly noisy examples. Then the programs could be evaluated on all examples and the program with the highest accuracy is kept.
> Further, the examples that are missclassified by the best performing program can be determined. Those examples are possibly faulty and this information could be used to optimize the object extractor. With this method the object extractor and program synthesis module could be optimized iteratively.
>
>
> **5. Extending Pix2Code to harder image settings.**
>
> Integrating the methods of Skryagin et al. [1] represents a well-suited candidate extension for this concern. Particularly, as it was originally introduced in the context of slot attention based object representations.
> Overall, however, we wish to emphasize the importance of maintaining symbol-like representations for a model's interpretability and revisability [4]. Real-valued attributes can, however, easily be processed by adapting the DSL to also operate on float values.
>
> **6. Example primitives.**
>
> Sure, thanks for your interest! In the following we provide a graph of all the abstracted primitives of a Pix2Code model trained on CURI iid: [here](https://anonymous.4open.science/r/pix2code-C445/figures/abstracted_primitives.png)
>
> Let us highlight a few things here. We observe that one of the basic primitives is
> `f0=(λ (x y) (forall (λ (z) (eq? y (index x z)))))`
> This represents a function that can be applied on a list of lists and checks for each element if the value at index `x` is equal to `y`. This is quite general and can be used to compare colors, shapes, etc.
>
> In another primitive, `f1=(f0 5)`, the primitive `f0` gets extended
> so that the parameter `x` gets set by the value `5` (which represents the index for color).
> `f1` is then integrated in other primitives, e.g.`f8=(f1 6)` and `f18=(f1 0)` where y is set respectively with the values `6` and `0` for the colors `purple` and `gray`. This means that `f8` resembles the concept *all objects are purple* and `f18` the concept *all objects are gray*.
>
> We have added the figure and a discussion therof to the appendix.
>
> [1] Skryagin, Arseny, et al. "Neural-probabilistic answer set programming." KR (2022).
>
> [2] Blondel, Mathieu, and Vincent Roulet. "The Elements of Differentiable Programming." arXiv (2024).
>
> [3] Skryagin, Arseny, et al. "Scalable neural-probabilistic answer set programming." Journal of Artificial Intelligence Research 78 (2023).
>
> [4] Kambhampati, Subbarao, et al. "Symbols as a lingua franca for bridging human-ai chasm for explainable and advisable ai systems." AAAI (2022).

---

### Official Review · Reviewer_DnVA · 2024-03-26

**Q2-1 Originality-Novelty:** 3
**Q2-2 Correctness-Technical Quality:** 3
**Q2-5 Clarity Of Writing:** 3

**Q1 Summary And Contributions:**

In this paper, a program synthesis method from pixel-level visual inputs is proposed. The proposed method has two functionalities: transforming visual inputs into abstract concepts, which utilizes the sequential visual tokenization method similar to pix2seq [1], and learning the program synthesis model by decomposing the model into learnable primitives and the code model. The proposed method is evaluated on benchmark visual program synthesis datasets and shows its effectiveness.

**Q2-3 Extent To Which Claims Are Supported By Evidence:**

3: Good: the main claims are supported by convincing evidence (in the form of adequate experimental evaluation, proofs, (pseudo-)code, references, assumptions).

**Q2-4 Reproducibility:**

3: Good: key resources (e.g. proofs, code, data) are available and key details (e.g. proofs, experimental setup) are sufficiently well-described for competent researchers to confidently reproduce the main results.

**Q3 Main Strengths:**

- The paper is well-motivated and address a significantly meaningful neuro-symbolic learning problem.

- In my view, the design of the system is technically sound. The proposed learning method for the high-level program, which is based on the wake-sleep algorithm.

**Q4 Main Weakness:**

- The visual perception model is trained independently from the program synthesis model, which is somehow not quite satisfactory. This makes the difference from the pure program synthesis task smaller.

**Q5 Detailed Comments To The Authors:**

- I suggest provide analysis on time costs of program synthesis. To my knowledge, one of the major drawback of many existing neuro-symbolic methods lies in the high time cost of searching on the possible programs.

- More details on the wake-sleep algorithm introduced in the paper would be beneficial, which can be put in the appendix.

- I wonder whether more baselines in the experiments can be considered, such as $\alpha$-ILP? In my view, the ILP methods can also address the task described in the paper. It would be meaningful to provide more in-depth comparisons on their pros and cons.

**Q9 Complying With Reviewing Instructions:**

Yes

---

> ### Author Rebuttal · Authors · 2024-04-05
>
> **1. The visual perception model is trained independently**
>
> We agree that investigating dependent, jointly trainable approaches is very important future work. Overall, however, it is still very much an open research topic whether to combine both object detection and abstract reasoning during training [1,2,3], particularly, as object detection and abstract reasoning are tasks that require different skills. For now the aim of our work is to show how it is possible to harness the benefits of program synthesis for the challenging domain of visual concept learning. For this we focus on the abilities of a pre-trained perception model.
>
> **2. Analysis on time costs**
>
> Yes, we agree, thanks for this useful insight. We have provided these in the following where we present the mean durations (in sec.) for the training of CURI-B and Pix2Code (and its sub-modules) on the iid data set of CURI:
>
>
> |          | CURI-B          | Pix2Code | *Object Extractor* | *Program Synthesis* |
> | -------- | --------         | -------- | -------- | -------- |
> | Duration | 1094.3+-122s     | 48575.7+-2338s     | 17247.3+-583s | 31328.3+-2404s |
>
> The training of CURI-B for 1000 steps takes on average 1094 seconds (\~18 minutes). Pix2Code was trained for 15 iterations which takes on average 48,575.5 seconds (\~13,5h). Indeed, this is a substantantially longer training time. However, we consider this to be a trade-off for the benefits of improved generalisability, interpretability and revisability.
>
> We have added our new evaluations and a discussion of these into the Appendix.
>
> **3. ILP baseline experiments**
>
> This is a good point, it is true that ILP systems have similarities with our method and that they can be used to find logic programs for complex visual scenes. However, there are three key points in which our method differs from ILP methods like alpha-ILP:
> 1. **Different concept learning setup.** In our work we propose a concept learning approach that learns how to discover new concepts which it has not seen during training. ILP approaches (on images), on the other hand, are usually designed to solve *one* complex visual pattern. E.g., in alpha-ILP the model solves four Kandinsky Patterns individually and during evaluation it only provides solutions for those concepts which it has seen during training. Thus, overall, the *task-based concept learning* setting is not applicable out-of-the-box for alpha-ILP and would require additional updates and efforts to allow for this.
> 2. **Large datasets required.** ILP methods such as alpha-ILP require substantially more training data. E.g., the smallest training set of [4] contains 2k examples of the same concept. In this work we focus on *few-shot* visual concept learning as provided by the benchmark CURI dataset where we show that Pix2Code operates well with only 25 examples per concept.
> 3. **Required background knowledge.** Pix2Code does not require any background knowledge besides the basic DSL. ILP methods, on the other hand, often require more background knowledge. E.g., alpha-ILP requires additional predicates like same_color, different_color etc. which represent relations that need to be pre-defined by the human. Providing these would represent a different (perhaps unfair) setting to our evaluations.
>
> Overall, we do agree that comparing Pix2Code to ILP baselines is an interesting comparison, however given the concerns above we consider this requires additional updates to current ILP models which is valuable future work.
>
> [1] Koh, Pang Wei, et al. "Concept bottleneck models." ICML (2020).
>
> [2] Yangyi Chen, Xingyao Wang, Manling Li, Derek Hoiem, Heng Ji:
> ViStruct: Visual Structural Knowledge Extraction via Curriculum Guided Code-Vision Representation. EMNLP (2023).
>
> [3] Hu, Ronghang, et al. "Learning to reason: End-to-end module networks for visual question answering." CVPR (2017).
>
> [4] Shindo, Hikaru, et al. "α ILP: thinking visual scenes as differentiable logic programs." Machine Learning 112.5 (2023).

---

### Meta-Review · Area_Chair_LMcN · 2024-04-15

**Summary:** The paper presents a framework for abstract visual concept learning by combining an object extractor (where each image is represented as a set of objects with interpretable discrete-valued attributes) with a lambda-calculus based program synthesis generator to derive a simple expression (program) that explains a split between positive and negative examples of a compositional concept. The method proposed is sound and performs well empirically (better than end-to-end trained systems), with the added benefit of being highly interpretable.

**Recommendation:** All reviewers are in favor of accepting the paper and so am I. The paper scores high or very high across all dimensions (novelty, clarity, quality, evidence, etc). Reviewers provided some valuable feedback, which has been addressed by the authors and should lead to an improved camera-ready manuscript. This paper is the highest-scoring in my batch (and I think the score represents the quality of the work) - I therefore recommend 'Accept as a Spotlight' (the final decision on this of course depends on the quality of all other submissions, over which I have very little visibility). The method is interesting and sound, with good results, and I think the work is interesting to a broader audience. Informally, I would not be surprised if the paper ended up in the **top 20% of accepted** papers, but since I only have visibility over a small batch of submissions I cannot really make a qualified statement.